# Virtual alignment of pathology image series for multi-gigapixel whole slide images

Chandler D. Gatenbee [1]✉, Ann-Marie Baker [2], Sandhya Prabhakaran [1], Ottilie Swinyard[2], Robbert J. C. Slebos [3], Gunjan Mandal[4], Eoghan Mulholland[5], Noemi Andor[1], Andriy Marusyk [6], Simon Leedham[5], Jose R. Conejo-Garcia [4], Christine H. Chung [3], Mark Robertson-Tessi [1], Trevor A. Graham [2] & Alexander R. A. Anderson [1]✉

Interest in spatial omics is on the rise, but generation of highly multiplexed images remains challenging, due to cost, expertise, methodical constraints, and access to technology. An alternative approach is to register collections of whole slide images (WSI), generating spatially aligned datasets. WSI registration is a two-part problem, the first being the alignment itself and the second the application of transformations to huge multi-gigapixel images. To address both challenges, we developed Virtual Alignment of pathoLogy Image Series (VALIS), software which enables generation of highly multiplexed images by aligning any number of brightfield and/or immunofluorescent WSI, the results of which can be saved in the ome.tiff format. Benchmarking using publicly available datasets indicates VALIS provides state-of-the-art accuracy in WSI registration and 3D reconstruction. Leveraging existing open-source software tools, VALIS is written in Python, providing a free, fast, scalable, robust, and easy-to-use pipeline for registering multi-gigapixel WSI, facilitating downstream spatial analyses.

Cellular interactions and the structure of the tumor microenvironment can affect tumor growth dynamics and response to treatment[1-3]. Interactions and the effect of tissue structure can be elucidated via spatial analyses of tumor biopsies, although there are many challenges. Among these are the limited number of markers that can be detected on a single tissue section. This can be overcome by repeated cycles of staining on the same tissue section or by staining serial slices for different subsets of markers. However, the captured images will likely not align spatially due to variance in tissue placement on the slide, tissue stretching/tearing/folding, and changes in physical structure from one slice to the next. Without accurate alignment, spatial analyses remain limited to the number of markers that can be detected in a single

section. While there are methods that can stain for a large number of markers on a single slide, they are often highly expensive, destructive, and require considerable technical expertise[4-8].

Image registration is the process of aligning one image to another such that they share the same coordinate system, and therefore offers the potential to align histology images. However, aligning histology images presents several challenges, which include: spatial variation in color intensity due to markers binding in different regions of the tissue; lack of a common marker across images (in the case of IHC); inter-user or inter-platform variation in staining intensity; tissue deformations (e.g., stretching, folds, tears); unknown order of serial sections; large numbers of images; and massive file sizes, often several gigabytes

[1]Department of Integrated Mathematical Oncology, H. Lee Moffitt Cancer Center & Research Institute, 12902 Magnolia Drive, SRB 4, Tampa, FL 336122, USA. [2]Evolution and Cancer Laboratory, Centre for Genomics and Computational Biology, Barts Cancer Institute, Queen Mary University of London, London EC1M 6BQ, UK. [3]Department of Head and Neck-Endocrine Oncology, H. Lee Moffitt Cancer Center & Research Institute, 12902 Magnolia Drive, CSB 6, Tampa, FL, USA. [4]Department of Immunology, H. Lee Moffitt Cancer Center & Research Institute, 12902 Magnolia Drive, MRC, Tampa, FL 336122, USA. [5]Wellcome Centre for Human Genetics, University of Oxford, Oxford OX37BN, UK. [6]Department of Cancer Physiology, H. Lee Moffitt Cancer Center & Research Institute, 12902 Magnolia Drive, SRB 4, Tampa, FL, USA. ✉e-mail: Chandler.Gatenbee@moffitt.org; Alexander.Anderson@moffitt.org

(GB) when uncompressed (Fig. 1). For these reasons, among others, registration of WSI remains a challenging problem.

There have been a great many efforts to develop computational methods that automatically register WSI, with some more recent methods shown in Table 1. Some are limited to hematoxylin and eosin (H&E) staining[9–12], while others are designed to work with slides stained for different markers[13–18]. Some are designed to align only 2 slides[19,20], while others can align multiple slides[9,21]. There also exist methods to register immunofluorescence (IF) images, which can be an easier task as each image usually contains a DAPI channel that stains for nuclei[22].

Table 1 describes the features present in several of the more recent WSI registration methods. We would argue that for a method to be scalable, it must: be able to read, warp, and write full resolution WSI to facilitate downstream analyses; be fully automated; have a command line interface so that the registrations can be performed on high-performance computer (HPC) clusters; be freely available to everyone. Examination of Table 1 reveals that only a handful of methods have the complete set of features that make a method scalable. However, those that do have the ability to scale tend to be designed for specific problems, such as: aligning only brightfield images[20,21,23–25]; aligning only images that have been stained with H&E[9]; designed to construct an image by registering raw image tiles, meaning it can't be applied to the stitched images that are already poorly aligned[26]. Importantly, each of these methods require the user to specify a reference image to which the others will be aligned.

Selecting the correct reference image is not trivial, especially when an H&E image is not available. Choice of a reference image can make or break the registration when applied to datasets with more than two images. This is because for these methods to work, one must determine which slide looks most like the others, and if that chosen slide is not similar enough to the rest, the registration can fail, as some may align to it, while others do not.

Here, we present Virtual Alignment of pathoLogy Image Series (VALIS), a fully automated, flexible, scalable, and robust WSI registration software package that works with any number of brightfield and/or immunofluorescence images, without the need to define a reference image. VALIS is also fully documented with examples on ReadTheDocs (https://valis.readthedocs.io/en/latest/), and available for download on GitHub (https://github.com/MathOnco/valis)[27], PyPi (https://pypi.org/project/valis-wsi/), and DockerHub (https://hub.docker.com/r/cdgatenbee/valis-wsi). VALIS therefore provides several features that make it a useful and practical solution for WSI registration:

1. A new groupwise, multi-resolution, multi-modal, rigid and/or non-rigid registration method that can align any number of images while solving the issue of needing to define a reference image.
2. Easy to use software that can register brightfield and/or immunofluorescence (IF) images, the latter of which can be merged into a single multi-channel image.
3. Registration software that can read 322 formats using Bio-Formats or Openslide, meaning it can be used with the majority of WSI[28,29].

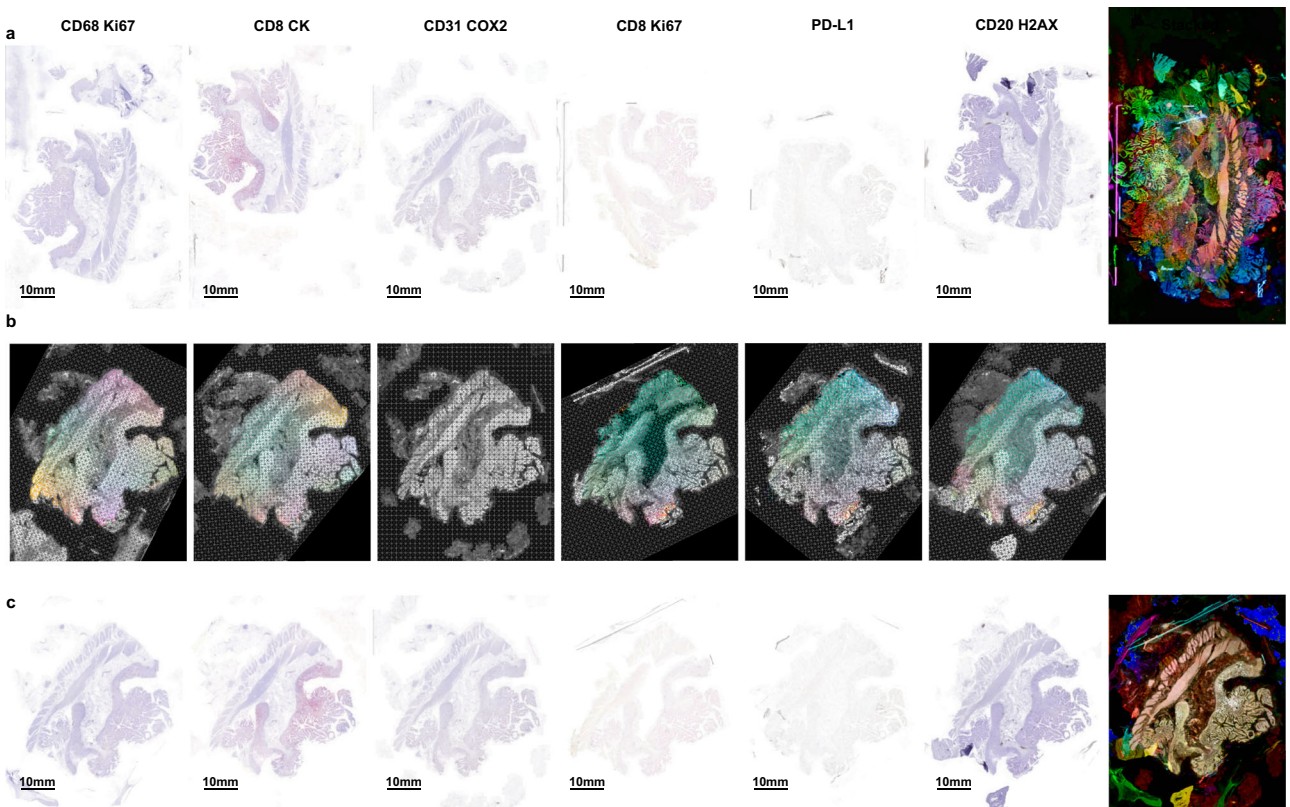

**Fig. 1 | Example of a challenging dataset.** VALIS handles potential batch effects from IHC images that would otherwise make image registration challenging. Such batch effects include large displacements (rotation, translation, etc.); deformations (stretches, tears); and spatial variation in color and luminosity due to differing spatial distributions of markers and/or different staining protocols. Large file sizes also present challenges to registering whole slide images (WSI). **a** Six serial slices of a colorectal adenoma were stained by three different individuals, with each marker stained with Fast Red or DAB. Note the substantial spatial variation in color and brightness, due to the heterogeneous spatial distribution of different cell types (each type stained with a different marker), and different staining protocols where some images are heavily stained and others lightly stained. The rightmost image shows the result of stacking the un-registered images, where each color shows the normalized inverted luminosity of each image. Each slide is also too large to open in memory, with each being ~32GB when uncompressed. **b** VALIS finds the rigid and non-rigid transformations to align the WSI. These images show the same transformation applied to a triangular mesh overlaid on each image, with color indicating the direction and magnitude of change. **c** Left: Alignment of the same slides using VALIS. Right: Image stack after image registration using VALIS. The transformations found by VALIS can subsequently be used warp each of the 32 Gb slides, which can be saved as ome.tiff images for downstream analyses.

**Table 1 | Features of recent approaches to register whole slide images**

| | Source data | | | | High Throughput Capability | | Transformations | | In/Out | | | | Accessibility | | Benchmarking | |
|---|---|---|---|---|---|---|---|---|---|---|---|---|---|---|---|---|
| | Brightfield | Different IHC stains | IF | >2 images | Fully automated | Command line interface | Rotation invariant | Non-rigid | N WSI formats | Works with stitched images | Warps and saves WSI as ome.tiff | Warps points | Available for download | Free | ANHIR | ACROBAT 2022 |
| VALIS (proposed) | ✓ | ✓ | ✓ | ✓ | ✓ | ✓ | ✓ | ✓ | 322 (BF OS) | ✓ | ✓ | ✓ | ✓ | ✓ | ✓ | ✓ |
| ASHLAR (2021)[22] | ✓ | ✓ | ✓ | ✓ | ✓ | ✓ | ✗ | ✓ | 319 (BF) | ✗ | ✓ | ✗ | ✓ | ✓ | ✗ | ✗ |
| DeepHistReg/AGH/AGHSSO (2022)[59] | ✓ | ✓ | ✗ | ? | ✓ | ✓ | ✓ | ✓ | ? | ✓ | ? | ✓ | ✓ | ✓ | ✓ | ✓ |
| MEVIS (2022)[60] | ✓ | ✓ | ✗ | ✗ | ✓ | — | ✓ | ✓ | — | ✓ | ✗ | ✓ | ✗ | — | ✓ | ✓ |
| NEMESIS (2022)[61] | ✓ | ✓ | ✗ | ✗ | ✓ | ✓ | ✓ | ✓ | ? | ✓ | ✗ | ✓ | ✓ | ✓ | ✗ | ✓ |
| MEDAL (2022) in ref. 62 | ✓ | ✓ | ✗ | ✗ | ✓ | ✓ | ✓ | ✓ | 10 (OS) | ✓ | ✗ | ✓ | ✓ | ✓ | ✗ | ✓ |
| TUB (2020) in ref. 33 | ✓ | ✓ | ✗ | ✗ | ✓ | — | ✓ | ✓ | — | ✓ | ✗ | ✓ | ✗ | — | ✓ | ✗ |
| TUNI (2020) in ref. 33 | ✓ | ✓ | ✗ | ✗ | ✓ | — | ✓ | ✓ | — | ✓ | ✗ | ✓ | ✗ | — | ✓ | ✗ |
| CKVST (2020) in ref. 33 | ✓ | ✓ | ✗ | ✗ | ✓ | — | ✓ | ✓ | — | ✓ | ✗ | ✓ | ✗ | — | ✓ | ✗ |
| Jiang et al.[23] | ✓ | ✓ | ✗ | ✗ | ✗ | ✓ | ✗ | ✗ | 10 (OS) | ✓ | ✗ | ✗ | ✓ | ✓ | ✗ | ✗ |
| Paknezhad et al.[21] | ✓ | ✓ | ✗ | ✗ | ✗ | ✓ | ✓ | ✗ | ? | ✓ | ✗ | ✗ | ✓ | ✓ | ✗ | ✗ |
| Liang et al.[24] | ✓ | ✓ | ✗ | ✓ | — | — | ✓ | ✓ | — | ✓ | — | — | ✗ | — | ✗ | ✗ |
| Marzahl et al.[25] | ✓ | ✓ | ✗ | ✗ | ✗ | ✓ | ✓ | ✓ | 10 (OS) | ✓ | ✗ | ✓ | ✓ | ✓ | ✗ | ✓ |
| Kajihara et al.[58] | ✓ | ✗ | ✗ | ✓ | — | — | ✓ | ✓ | — | ✓ | — | ✗ | ✗ | — | ✗ | ✗ |
| HistoReg (2021)[20] | ✓ | ✓ | ✗ | ✗ | ✓ | ✓ | ✓ | ✓ | 4 | ✓ | ✗ | ✓ | ✓ | ✓ | ✓ | ✗ |
| SFG (2022)[34] | ✓ | ✓ | ✗ | ✗ | ✓ | ✓ | ✓ | ✓ | — | ✓ | ✗ | ✓ | ✓ | ✓ | ✓ | ✗ |
| CODA (2022)[19] | ✓ | ✗ | ✗ | ✓ | ? | ? | ✓ | ✓ | ? | ✓ | ? | ✗ | — | — | ✗ | ✗ |
| HALO | ✓ | ✓ | ✓ | ? | ? | ? | ✓ | ✓ | ? | ✓ | ✓ | ? | ✓ | ✗ | ✗ | ✗ |
| Visiopharm | ✓ | ✓ | ✓ | ? | ? | ? | ✓ | ✓ | ? | ✓ | ✓ | ? | ✓ | ✗ | ✗ | ✗ |

"✓" indicate the feature is present, "✗" indicates the feature is absent, "—" indicates N/A, and "?" indicates unknown, the latter of two which may occur if the method/software is currently unpublished and/or unavailable for download. Check marks in the *Brightfield* column indicate that the software can register brightfield images, such as those stained using traditional immunohistochemistry. *Different IHC stains* indicate whether or not the method supports images with multiple stains (i.e., not all images must have H&E). The *IF* column describes whether or not the method can register images stained using immunofluorescent probes. The *>2 images* column indicates whether or not the method was designed to register more than two images. *Rotation invariant* indicates whether or not the registration method can account for rotations. The *Non-rigid* column indicates whether or not the method can perform non-rigid registration. *Fully automated* means that the user needs only to provide the minimal amount of information (such as source and destination directories) to open, register, and save the WSI. *Command line interface* indicates whether or not the method/software can be called from the command line. The *N supported formats* column describes the number of image formats the software can read, which is determined by the readers used by the software: BF = Bio-Formats (319 formats, all IHC, 3 of which cannot be read by Bio-Formats (.mrxs,.svslide,.vmu)). *Works with stitched images* indicates that the software is able to register slides that have already been stitched, i.e., saved as a single file. *Saves WSI* denotes whether or not the software can write the registered WSI to file in the ome.tiff format, allowing registered image could be further processed downstream. Check marks in *Warps points* mean that the method/software natively provides the ability to warp points, such as cell positions or annotations, using the registration parameters. *Available for download* indicates that the software is readily available, on Github, PyPi, conda-forge, commercial site, etc.. *Opensource and free* indicates that the software (if available) is opensource and free. *Benchmarking* indicates performance measured using publicly available datasets, in this case the ANHIR and ACROBAT 2022 grand challenges.

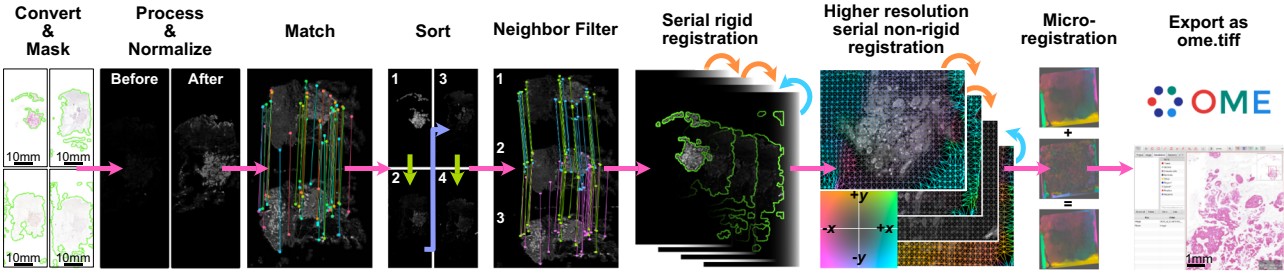

**Fig. 2 | Overview of the VALIS alignment pipeline.** VALIS uses Bio-Formats and OpenSlide to read the slides and convert them to images for use in the pipeline, meaning it is compatible with 322 image formats. Once converted from slides, masks are created and applied to each image, which will focus the registration on the tissue (mask outline highlighted in green). Next, images are processed and normalized to look as similar as possible. Features are then detected in each image and then matched between all possible pairwise combinations. Feature distances are used to construct a distance matrix, which is then clustered and sorted, ordering the images such that each image should be adjacent to its most similar image. Once ordered, the matches undergo another round of filtering, wherein features used to rigidly register an image to the next image must also have been matched in the previous image. Unique matches to previous and next images are shown as purple or blue lines, while those remaining after neighbor filtering are shown as green lines. Images are then aligned serially towards the center of the image stack (reference image, if specified), going from the inside out. Serial rigid transformations are found first, using the neighbor-filtered matches. The bounding box around the region where masks overlap or touch is used to slice out a higher resolution image for use in non-rigid registration, again performed serially from the inside out. The non-rigid transformations are accumulated as one moves towards the edges of the stack, which can bring distant features together. A final optional "micro-registration" step can be performed, which is accomplished by performing a second non-rigid registration on higher resolution non-rigidly warped images. Once registration is complete, the slides can be warped and saved in their native resolution as ome.tiff images for downstream analyses.

4. Provides the ability to warp, save, and convert huge multi-gigapixel WSI to ome.tiff, an opensource image format that can store multiple large pyramid images with metadata[28–32].

5. Offers a straight-forward way to warp coordinates using image transformation parameters. Can therefore be used to warp cell coordinates from existing cell segmentation data, transfer annotations from one image to another, etc...

6. Software that can also warp WSI with user-provided transformations and/or registration methods (by sub-classing the relevant object classes).

## Results

### Overview

A full description of the registration pipeline is described in the Methods section, but briefly, it goes as follows (Fig. 2):

1. WSI are converted to numpy arrays. As WSI are often too large to fit into memory, these images are usually lower resolution images from different pyramid levels.

2. Masks are created and applied to the images, focusing registration on the tissue.

3. Images are processed to single channel images. They are then normalized to one another to make them look as similar as possible.

4. Image features are detected and then matched between all pairs of images.

5. If the order of images is unknown, they will be optimally ordered based on their feature similarity, such that the most similar images are adjacent to one another in the z-stack. This increases the chances of successful registration because each image will be aligned to one that looks very similar. This step can be skipped if the order is known (such as with 3D tissue reconstruction).

6. Rigid registration is performed serially, with each image being rigidly aligned towards (or optionally, directly to) the reference image. That is, if the reference image is the 5th in the stack, image 4 will be aligned to 5 (the reference), and then 3 will be aligned to the now registered version of 4, and so on. If a reference image is unspecified, it will be set to be the image at the center of the z-stack. Only features found in both neighboring images are used to align the image to the next one in the stack. VALIS uses feature detection to match and align images, but one can optionally perform a final step that maximizes the mutual information between each pair of images.

7. The masks are rigidly warped and combined to create a non-rigid registration mask. The bounding box of this mask is then used to extract higher resolution versions of the tissue from each slide. The higher resolution images are then re-processed and used for non-rigid registration, which is performed either by: aligning each image towards (or to) the reference image following the same sequence used during rigid registration (the default); using groupwise registration that non-rigidly aligns the images to a common frame of reference.

8. One can optionally perform a second non-rigid registration using even higher resolution versions of each image. This is intended to better align micro-features not visible in the lower resolution images used in the previous steps, and so is referred to as "micro-registration".

9. Error is estimated by calculating the distance between registered matched features in the full resolution images.

The transformations found by VALIS can then be used to warp the full resolution WSI. It is also possible to merge non-RGB registered slides to create a highly multiplexed image. These aligned and/or merged slides can then be saved as ome.tiff images. The transformations can also be used to warp processed versions of the WSI (e.g., those that underwent color deconvolution, additional cleaning, etc...), or point data, such as cell centroids, annotations, polygon vertices, etc...

### Benchmarking

There exist many excellent registration methods (Table 1), so a key question is how well does VALIS perform in comparison? To address this question, we first benchmarked VALIS using the Automatic Non-rigid Histological Image Registration (ANHIR) Grand Challenge dataset[33]. This includes eight datasets of brightfield images originating from different tissues, with multiple samples and stains per dataset. Each image is accompanied by hand selected tissue landmarks that are evenly distributed across the image and found in other serial slices taken from the same sample, making it possible to measure biologically meaningful alignment accuracy between pairs of images. In total, there are 49 unique samples, with public training landmarks available for 230 image pairs. There are an additional 251 private landmark pairs used to evaluate registration accuracy after the user has uploaded their results. Therefore, the results presented on the leaderboard may differ slightly than what the user calculates using the publicly available training landmarks. The goal of the challenge is to register pairs of

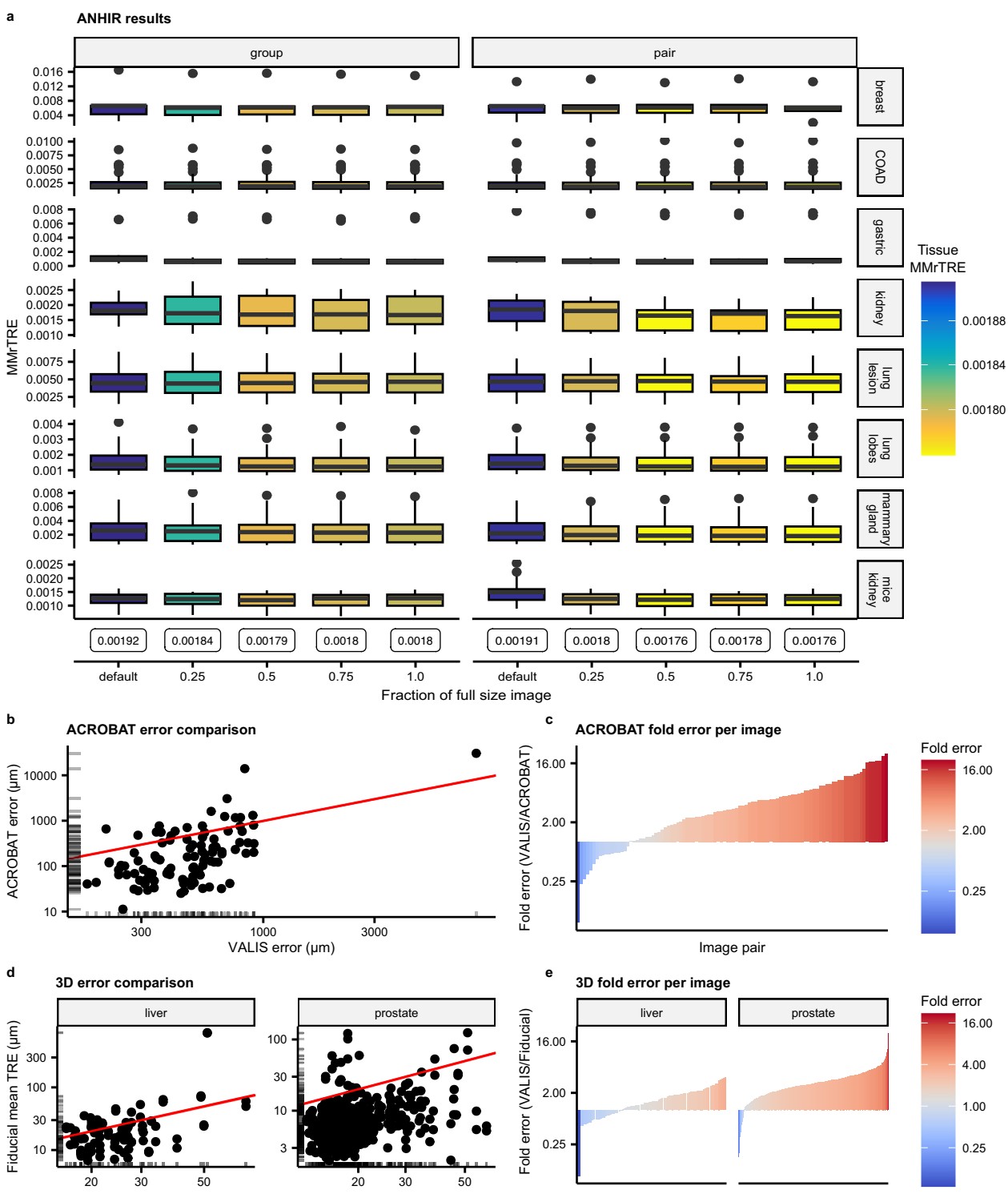

images, the accuracy of which can be measured by applying the transformations to the landmarks and then measuring the distance between the newly registered landmarks. More specifically, error is measured as the median relative target registration error (median rTRE), where rTRE is the Euclidean distance between registered landmarks, divided by the diagonal of the target image, thereby normalizing error between 0 and 1.

In addition to benchmarking VALIS using default parameters (e.g., groupwise registration using the default image sizes and no micro-registration), we also assessed performance using micro-registration, both using the groupwise approach (referred to as "group" in Fig. 3a),

or registering directly to the target image after an initial groupwise registration (referred to as "pair" in Fig. 3a). These experiments were conducted by performing micro-registration at four different levels of downsampling: 0.25, 0.5, 0.75, 1.0 (i.e., the full resolution image). The results of these experiments can be found in Fig. 3a.

VALIS' median median rTRE (MMrTRE, the primary metric used to rank scores on the leaderboard) is 0.00172, ranking 8/44 overall (although several methods have multiple entries), but being the second most accurate published and/or available method (as of February 20, 2023). A detailed breakdown of MMrTRE per tissue can be found in Supplementary Fig. 1. The majority of methods that rank higher are

**Fig. 3 | Benchmarking results. a** Benchmarking results of VALIS using the publicly available Automatic Non-rigid Histological Image Registration (ANHIR) Grand Challenge dataset. Values are the median median rTRE (MMrTRE) for each of the $N = 230$ unique image pairs used to assess registration accuracy. Each major column is for a registration strategy, with "group" meaning only groupwise registration was performed, while "pair" means that micro-registration was used to directly align each image to the target image after the initial groupwise registration. Minor columns indicate the amount of downsampling used for the micro-registration. Values inside the bubbles at the bottom of each minor column indicate the MMrTRE for all image pairs, the default metric used to rank methods on the ANHIR leaderboard. Rows indicate the tissue type. In each box, the center line indicates the median, the top and bottom indicate the 75th and 25th percentiles, respectively, the top whisker the largest value that is no further than 1.5 Interquartile range (IQR) from the 75th percentile, the bottom whisker the smallest value no more than 1.5IQR from the 25th percentile, and points indicate outliers. **b** VALIS' estimated performance (x-axis) versus actual performance (y-axis) in the ACROBAT Grand Challenge, with the solid line being the identity line. These results show that VALIS performed well in the grand challenge, but also that VALIS tends to overestimate error. **c** Waterfall plot showing the estimated error over the true error, illustrating that VALIS frequently overestimates error. **d** Using the Kartasalo (2018) dataset, error (i.e., mean TRE, in μm) calculated using the fiducial landmarks is compared to the error estimated by VALIS, which is measured using matched image features. Solid line is the identity line. These results show that VALIS tends to overestimate registration error, as the majority of points fall below the identity line (i.e., VALIS error > fiducial error). **e** Waterfall plot showing the estimated error (VALIS) divided by the true error (Fiducial), again showing that VALIS frequently overestimates registration error. Source data are provided as a Source Data file[57].

unpublished/unavailable, meaning one cannot use those methods. The exception is the Structural Feature Guided (SFG) convolutional neural network method, which uses sparse and dense structural components (e.g., SIFT features and pixel level feature maps, respectively) to optimize the structural similarity between the source and target image[34]. The goal of SFG is to provide an automated and highly accurate method to register histology images, which it does quite well. However, while SFG successfully addresses the first challenge of WSI registration, it does not address the second practical, but critical, issue of WSI registration, i.e., being able to read, warp, and write the registered multi-gigapixel images saved in specialized formats.

We next benchmarked VALIS using the The AutomatiC Registration Of Breast cAncer Tissue (ACROBAT) grand challenge dataset, which was part of MICCAI 2022[35]. Similar to ANHIR, the goal is to automatically register pairs of WSI, using hand-matched landmarks to quantify registration accuracy. However, unlike ANHIR, the images were collected from routine diagnostic workflows, and so often contained artifacts common to real world datasets, such as cracks, streaks, pen marks, bubbles, etc… that increased the difficulty of image registration. There are a total of 397 unique samples used to assess registration accuracy, with 100 images pairs used for a validation leaderboard (used to aid in developing the algorithm), and 297 for a test leaderboard, for which a total of 13,130 landmark pairs were used to assess accuracy. Importantly this means that only the images can be used to find the registration parameters. Scores are calculated by uploading the registered landmarks to the submission system. The primary score used to measure accuracy was the 90th percentile of physical distances between registered moving and fixed landmarks, in μm.

Due to the challenging nature of the images, a custom Image-Processer class was created to clean up and process the WSI. All image pairs were then registered using the default settings, followed by micro-registration using an image that was 25% of the full WSI's size. Using this approach, VALIS had a high accuracy score of 131.96 μm in the validation dataset, and 123.3 μm in the test dataset (y-axis of Fig. 3b).

As ACROBAT measures error in μm, and VALIS estimates error based on matched features, this dataset makes it possible to determine how well VALIS estimates error. It should be noted that prior to VALIS' error calculation, the matching feature positions used to calculate error are scaled to their location in the full resolution image, and then converted to biological units, if possible. Figure 3c shows the relationship between the estimated (VALIS) and true (ACROBAT) errors, with VALIS estimated error on the x-axis, error based on ACROBAT's hand annotations on the y-axis, and the identity line in red. These results indicate that VALIS tends to overestimate the error, with the true error being much lower. This discrepancy may be because the features used by VALIS to estimate error are based on much smaller versions of the WSI, and so their position is not as precise as those detected by hand.

3D reconstruction is also a common use case for WSI registration. To assess the performance of VALIS with this task, we performed benchmarking using the datasets described by ref. 36. Briefly, two datasets are provided: one murine prostate to be reconstructed from 260 serially sliced 20x H&E images (0.46 μm/pixel, 5 μm in depth), and one murine liver to be reconstructed from 47 serial slices (0.46 μm/pixel, 5 μm in depth). Accuracy of the alignment of the liver is determined using the positions of laser-cut holes that pass through the whole tissue, and should, in theory, form a straight line. In the case of the prostate dataset, for each pair of images, human operators determined the location of structures visible on both slices, preferentially selecting nuclei split by the sectioning blade. The authors refer to these landmarks as "fiducial points", and TRE was measured as the physical distance (μm) between said fiducial points. We next compare VALIS' TRE to the methods presented in Tables 1 and 2 of ref. 36., which provides the mean TRE using observed landmarks (i.e., the "TRE μ" column). Benchmarking using the liver dataset indicates that VALIS produces a mean TRE of 52.98, compared to the baseline reference value of 27.3 (LS 1) (y-axis of Fig. 3d). In the case of prostate, VALIS scored 11.41 μm, compared to the baseline reference value of 15.6 (LS 1) (y-axis of Figs. 3d and 6e). According to the authors, methods with scores approaching the LS 1 value can be considered "highly accurate", indicating that VALIS is suitable for 3D reconstruction.

As with ACROBAT, the pixel unit, in μm, was provided by[36], again making it possible to compare the true registration error (as measured using the fiducial points) with the error estimated by VALIS, which is based on matching image features (Fig. 3d,e). Consistent with the ACROBAT dataset, this comparison indicates that VALIS tends to overestimate the error. As such, the error estimates produced by VALIS may serve better to provide a sense of successful registration and/or assessing how changing parameters affects the registration quality.

## Validation

To test the robustness and generalizability of VALIS, we performed image registration on an additional 613 samples, with images captured under a wide variety of conditions (Fig. 4). These images were collected for routine analysis, and so were not curated in any sort of way. That is, these are "real world" images that reflect the regular challenges associated with WSI registration. Each sample had between 2 and 69 images; 273 were stained using immunohistochemistry (IHC), and 340 using immunofluorescence (IF); 333 were regions of interest (ROI) or cores from tumor microarrays (TMA), while 280 were whole slide images (WSI); the original image dimensions ranged from $2656 \times 2656$ to $104,568 \times 234,042$ pixels in width and height; 162 underwent stain/wash cycles, 451 were serial slices; 49 came from breast tumors, 109 from colorectal tumors, 156 from head and neck squamous cell carcinomas (HNSCC), and 299 from ovarian tumors. In total, this validation dataset involved registering and calculating the error for 4099 image pairs.

For the most part, the default parameters and methods were used to align the images in all of the datasets. The exception was how much to resize the images. Typically, full slides were resized to have a maximum width or height around 850–1000 pixels, while smaller images with cellular resolution (e.g., TMA cores, ROI) were resized to have maximum width or height around 500 pixels.

For each image, registration error was calculated as the median distance (μm) between the features in the image and the corresponding matched features in the neighboring image (see Methods section for more details) (Fig. 5). The registration error of the sample was then calculated as the average of the images' registration errors, weighted by the number of matched features per pair of images. The registrations provided by VALIS substantially improved the alignments between images, particularly in the case of serial IHC (Fig. 5a). As suggested in Fig. 3b-e, the quality of the registration may actually be better than the estimated error suggests.

### Potential applications

We next provide a few examples of how the registration parameters found by VALIS can be used to facilitate spatial analyses (Fig. 6). One strategy is to use the registration parameters to align and merge the full resolution WSI to create highly multiplexed images. Figure 6a shows the results of registering and merging 11 rounds of CyCIF to create a single 32 channel image. A similar approach can be taken for brightfield images as well, as shown in Fig. 6b, where VALIS was used to align 18 cyclically stained IHC images. The registration parameters were then used to warp and merge stain-segmented versions of each WSI, thereby creating single 18-channel image from cyclical IHC staining. In these cases, the merged CyCIF image was accurate enough for spatial analyses using cell positions, while quadrat-based approaches could be used to spatially analyze the less accurate brightfield IHC alignments (see Supplementary information and Supplementary Fig. 4). Both analyses highlight how the multiplexed images generated by VALIS can be used to quantify the tumor ecology, shedding light the role of the microenvironment and quantifying spatial interactions.

Cell segmentation can be a difficult task, and it may not be desirable to re-perform that analysis on newly registered images. For this reason, VALIS also provides the functionality to warp coordinates, meaning that the registration parameters can be used to warp existing cell segmentation data (Fig. 6c). These methods could also be used to warp annotations and regions of interest coordinates. Since VALIS can register different modalities (Fig. 6d), this also makes it possible to transfer annotations from H&E images to a corresponding IF dataset. As discussed above, VALIS can also be used for 3D tissue reconstruction (Fig. 6e). Other promising areas where VALIS, and image registration in general, could be useful are, among others, aiding generation of training data for virtual staining[37], or aligning spatial transcriptomics images to brightfield and/or IF images.

### Discussion

VALIS introduces a new groupwise WSI rigid and/or non-rigid registration method that increases the chances of successful registration by ordering images such that each will be aligned to their most similar image. This is performed serially, meaning that transformations accumulate, aligning all images towards the center of the image stack or a specified reference image. Since images are aligned as a group, there is no need to select a reference image, which can make or break a registration. There is an emphasis on acquiring good rigid registration, which in turn facilitates accurate non-rigid registration. This is accomplished through automated pre-processing, normalization, tissue masking, and three rounds of feature match filtering (RANSAC, Tukey's approach, and neighbor match filtering) (Fig. 2). VALIS also provides the option to refine the registration through micro-registration using a higher-resolution image. Finally, VALIS is flexible, being able to register both brightfield and fluorescence images. Combined with the selection of feature detectors, feature descriptors,

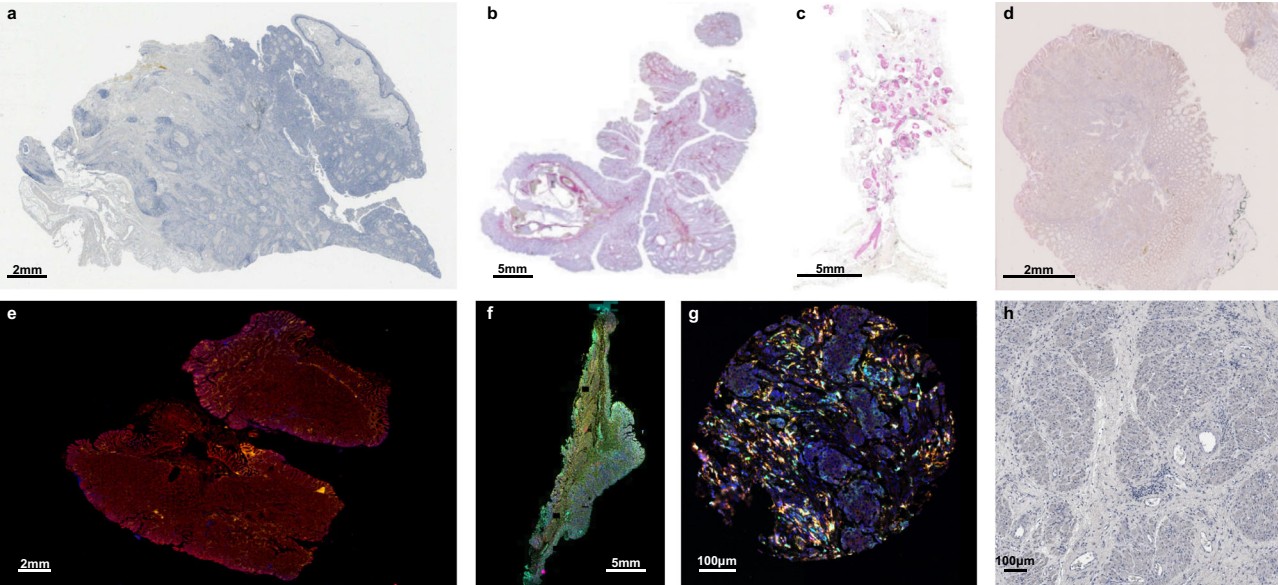

**Fig. 4 | Example of images of extra datasets used to validate VALIS. a** Head and neck squamous cell carcinoma (HNSCC). This sample set included four marker panels, each of which included between 13–20 markers stained using IHC. A single slice underwent the corresponding number of stain wash cycles, but all 69 images collected from the four panels have also been co-registered. **b** Human colorectal carcinoma or adenoma IHC serial slices, each with 1–2 markers per slide, and 6 slides per sample. **c** DCIS and invasive breast cancer serial slices, 1–2 markers per slide (stained using IHC), seven slides per sample. **d** Human colorectal carcinomas and adenomas, stained using RNAscope, 1–2 markers per slide, five slides per sample. **e** Human colorectal carcinomas and adenomas, stained using cyclic immunofluorescence (CyCIF), 11–12 images per sample. **f** Human colorectal carcinomas and adenomas stained using immunofluorescence, two slides per sample. **g** In addition to registering WSI, VALIS can also be used to register images with cellular resolution, such as cores from an immunofluorescent tumor microarray (TMA) taken from human ovarian cancers (two slides per sample), or **h** 40× regions of interest from HNSCC samples, taken from images in the same dataset in (a).

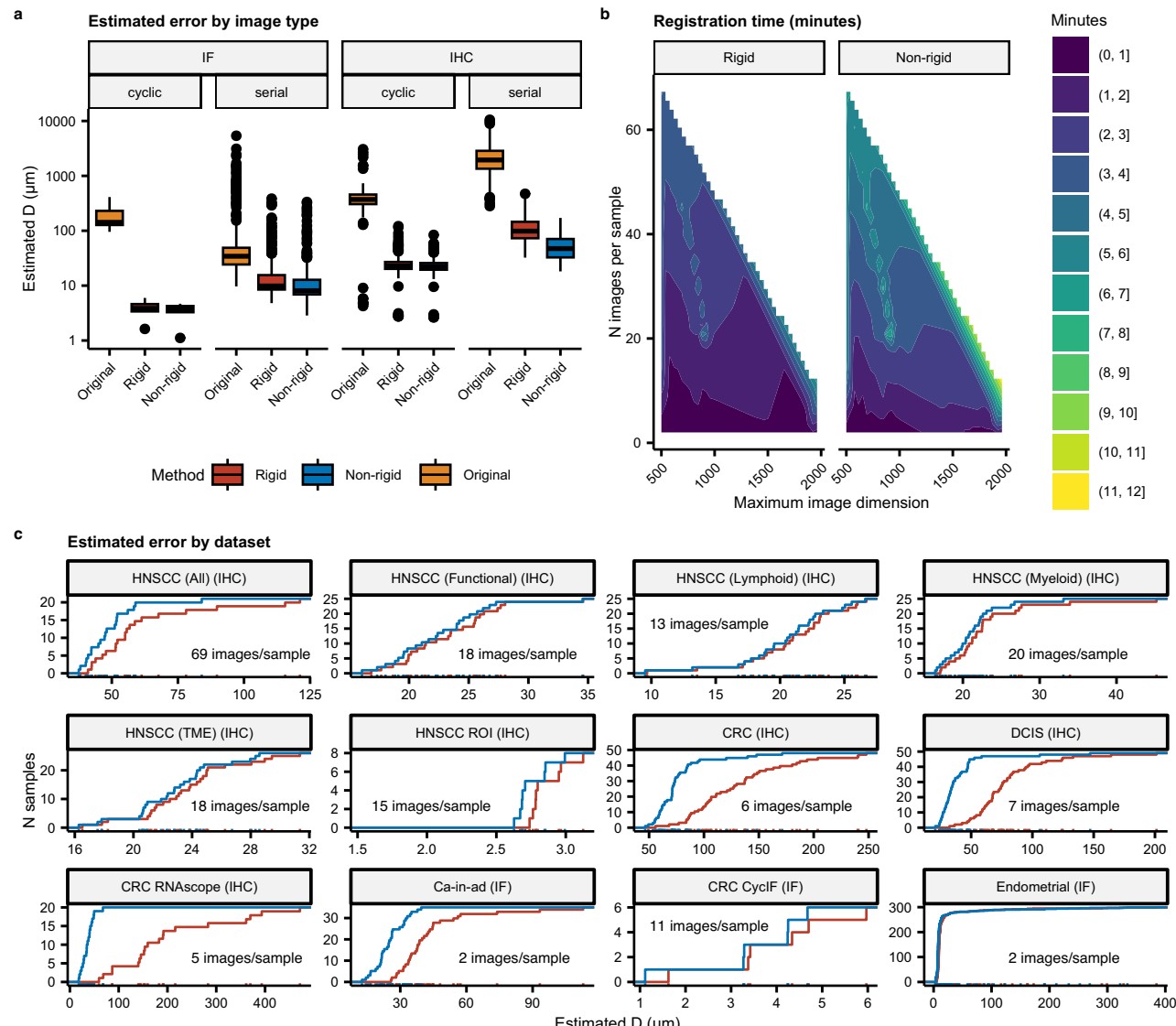

**Fig. 5 | Validation results.** Results of registering $N = 4099$ unique image pairs from the datasets shown in Fig. 4, which were captured from a variety of tissues, protocols, imaging modalities, and resolutions. **a** Boxplots showing the distance (µm) between matched features in the full resolution slides, before registration (yellow), after rigid registration (red), and then non-rigid registration (blue). In each box, the center line indicates the median, the top and bottom indicate the 75th and 25th percentiles, respectively, the top whisker has the largest value that is no further than 1.5 Interquartile range (IQR) from the 75th percentile, the bottom whisker the smallest value no more than 1.5IQR from the 25th percentile, and points indicate outliers. **b** Median amount of time (minutes) taken to complete registration as a function of the processed images' size (by largest dimension, on the x-axis) and the number of images being registered. These timings include opening/converting slides, pre-processing, and intensity normalization. **c** Empirical cumulative distribution plots of registration error for each image dataset. Source data are provided as a Source Data file[57].

and non-rigid registration methods, this approach is able to provide registrations with state-of-the-art accuracy.

In addition to other applications that can benefit from image registration (e.g., using annotations across multiple images, retrieving regions of interest in multiple images, warping cell coordinates, etc...), VALIS also makes it possible to construct highly multiplexed images from collections multi-gigapixel IF and/or IHC WSI. However, a challenge of constructing multiplexed images by combining stain/wash cycling and image registration is that multiple cycles eventually degrade the tissue and staining quality can decrease because antibody binding weakens as the number of cycles increases. It has been estimated that t-CyCIF images can reliably undergo 8-10 stain/wash cycles, and possibly up to 20 in some cases[38]. In our examples, we use four markers per cycle (including DAPI), suggesting one could construct a 24–60 plex image using a similar protocol. We suspect this number will increase as technological advances in staining protocols are made.

To maximize the number of markers used to generate multiplexed images with image registration, one can conduct experiments wherein each individual antibody undergoes multiple stain/wash cycles, measuring antibody sensitivity after each repeat. One can then use the results of these experiments to determine how many cycles an antibody can undergo while maintaining sensitivity. With this data in hand, one can order the staining sequence such that weaker stains are used first, and more robust stains are used in the later cycles. Such an approach should help maximize the number markers one can stain for. Using this approach, we were able to find staining sequences that allowed between 13–20 IHC stain/wash cycles per tissue slice (HNSCC data in Fig. 5c).

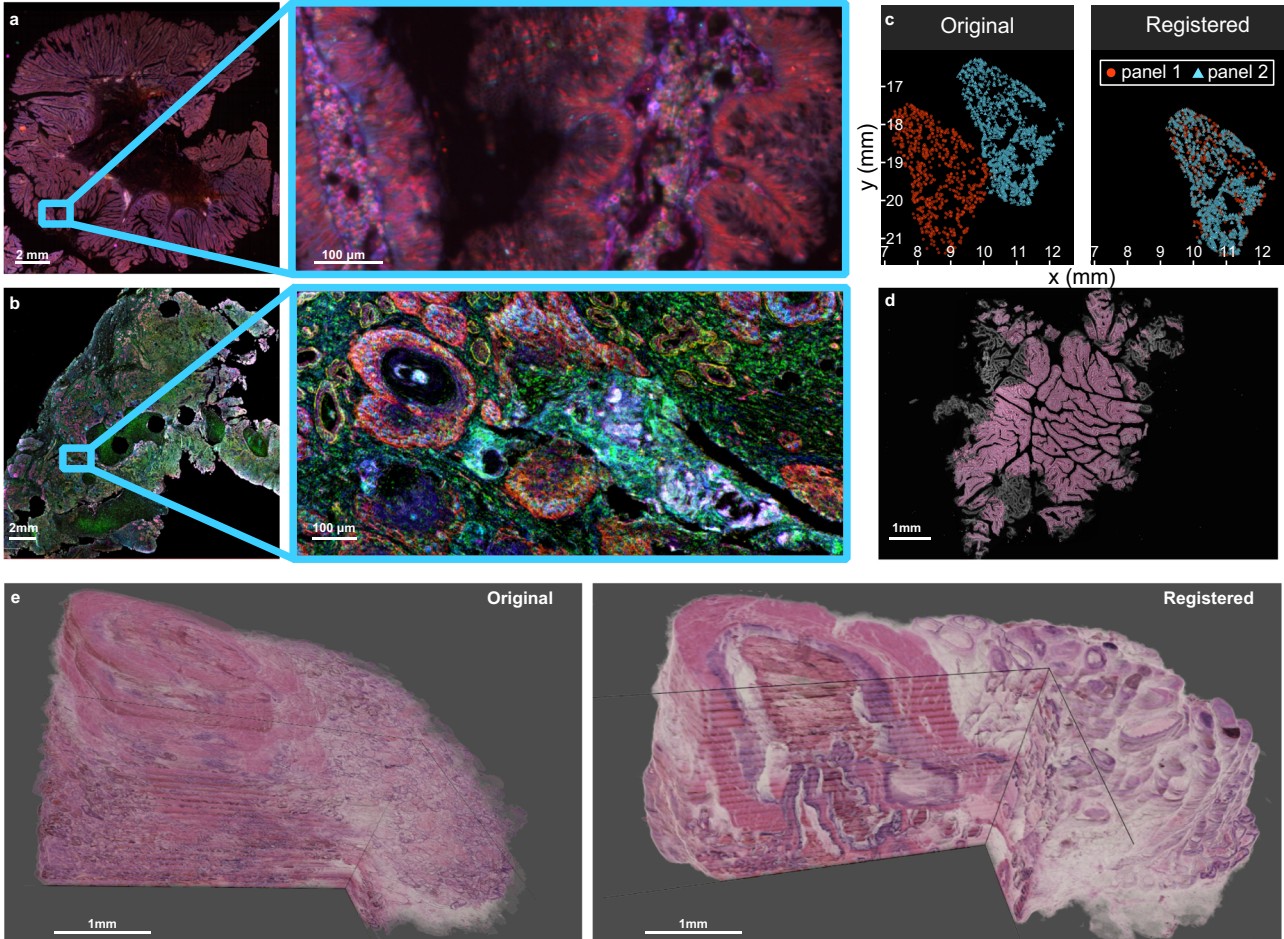

**Fig. 6 | Potential applications of VALIS. a** Merging registered CyCIF slides, in this case creating a 32-channel image. **b** Merging registered and processed IHC slides. Here, VALIS found the transformation parameters using the original images, but applied the transformations to stain segmented versions of 18 cyclically stained slides (see supplementary Table 3 for list of markers). **c** Applying transformations to cell segmentation data. **d** Registering an H&E slide to the DAPI channel of an IF slide, which may be useful in cases where annotations on H&E images would like to be used with IF images. Here, to visualize the alignment, the registered H&E image is overlaid on the DAPI channel. **e** 3D tissue reconstruction from serial sections. The left image shows the results of stacking the original images, while the right image shows the result of stacking the registered serial slices.

The impacts of stain/wash cycles on stain quality does not appear to affect registration accuracy. The CyCIF images, which underwent 11 cycles, had very low registration error, the maximum being estimated at 6μm. Likewise, most samples in the HNSCC panels had similar error distributions, despite each undergoing differing numbers of IHC staining rounds, being between 13 and 20 rounds depending on the staining panel (HNSCC data in Fig. 5c). These results did not include the micro-registration step, which should bring the error even lower. These experiments suggest stain/wash cycling has little impact on registration accuracy, and that VALIS is able to "fix" tissue deformations that can occur during repeated washing.

Whilst VALIS was designed to be robust, flexible, and easy to use, it does have several limitations. We have successfully tested VALIS on more than 5000 images, however, there will still be cases where registration fails. Such failures could be due to large distances between serial sections, severe tissue deformations, poor staining/lighting, major image artifacts, etc... Relatedly, while VALIS can register both RGB (i.e., brightfield images) and single/multi-channel images, it has only been tested with brightfield and immunofluorescent images. However, it is possible for a user to develop custom Python classes that can pre-process other image modalities for registration. These custom classes can then be used in the VALIS pipeline, thus making it possible to register additional modalities.

As VALIS is limited to the CPU, choosing to perform registration on very large images can be time-consuming (Supplementary Fig. 2c). This would mostly be relevant during the (optional) micro-registration step. It is worth noting, however, that accuracy gains acquired by registering the images at higher resolution tend to be small, and may or may not be worth the additional computational cost, depending on the application (Supplementary Fig. 2c).

Another limitation of VALIS is that it does not precisely estimate the accuracy of the registration (Fig. 3b-e). Perhaps fortunately, the error tends to be over-estimated, and the registration may be more accurate than the summary statistics suggest. An accurate judgment of the alignment is thus best made by the user.

Despite VALIS' limitations, and that it uses existing feature detectors, feature descriptors, and non-rigid registration methods, VALIS does provide several strengths. The approach of using feature matches to create and sort an image similarity matrix enables a pipeline that increases registration accuracy and has several additional benefits:

1. Sorting the images and then aligning serially ensures that images are aligned to the most similar looking image, increasing the chances of successful registration for each image pair.
2. Feature match quality is improved by using only the matches found in both neighbors, which should represent true tissue

features. This reduces the number of poor matches included in the estimation of rigid registration parameters, thereby increasing registration accuracy. This is only possible because the images have been sorted by similarity, and so an image can access the shared features of its neighbors.

3. Ordering the images and aligning them serially solves the non-trivial problem of selecting a reference image. If aligning more than two images, selecting the wrong reference image can cause a registration to fail. This can be especially challenging if an H&E image is not available, since it may not be obvious which image looks most similar to the others.

4. Because transformations accumulate, distant features in dissimilar images can be better aligned than might be possible if only performing pairwise registration. This too is only possible because the images have been sorted and aligned serially (Supplementary Fig. 3b).

5. Since images are sorted and aligned serially, any number of images can be registered at the same time, in contrast to manually conducting several pairwise alignments. Again, this is only possible because images were ordered by similarity.

In addition to presenting a new groupwise registration method, the preprocessing method described here is stain agnostic. Instead of extracting stains through color deconvolution (which requires providing or estimating a stain matrix, a challenge in itself), our method's approach is to standardize each image's colorfulness/chroma. This allows VALIS to work with a wide variety of stains.

Another strength of VALIS is that it provides a bridge between Bio-formats and libvips, making it easier to work with huge multi-gigapixel WSI saved in various formats. This interface is also available to the user, which we hope will help others in their projects involving very large WSI saved in specialized formats.

VALIS is also designed such that it is possible to add new image processors, feature detectors, descriptors, and non-rigid registration methods, meaning that it can be updated and improved over time, not being limited to the current choices or imaging modalities. As such, VALIS offers more than just access to a new WSI registration method, and we hope it will aid others with their work using WSI. In that vein, issues and feature requests can be handled through GitHub, such that VALIS can grow and accommodate the scientific image analysis community's needs.

## Methods
### Datasets
FFPE samples representing human adenomas, adenomas with foci of cancer ("ca-in-ads") and carcinomas (CRC) were selected from the histopathology archives of University College Hospital, London, under UK ethical approval (07/Q1604/17) or John Radcliffe Hospital, Oxford under ethical approval (10/H0604/72). DCIS samples originated from the biobank at University College Hospital, London. Ovarian tumor tissue microarrays (TMAs) were obtained from three different resources: the Tissue Core at Moffitt Cancer Center in Tampa, FL (approval MCC no. 50264); TriStar Technology Group, LLC (Rockville, MD); and US BioMax, Inc. (Derwood, MD). HNSCC samples were collected at Moffitt Cancer Center in Tampa, FL (approval MCC no. 18754). Datasets for the ANHIR grand challenge are described in[33], while the ACROBAT dataset is described in[35]. The datasets used to test 3D tissue reconstruction are described in[36]. Written informed consent was waived by the relevant RECs due to the retrospective and anonymous nature of this study.

### Reading the slides
Whole slide images (WSI) can be challenging to work with because they are frequently saved using various formats and are often several gigapixels in size (i.e., several billions of pixels). The resulting uncompressed files are frequently on the order of 20GB in size, which often precludes opening the entire image directly in memory. To address the issues of working with WSI, VALIS uses libvips (with OpenSlide support) and Bio-Formats[28,29] to read each slide. In cases where the format is not supported by libvips/OpenSlide, VALIS uses Bio-Formats to read the image in tiles, converting those tiles to libvips images, and then combining the tiles to rebuild the entire image as single whole-slide libvips image[29,32]. The images used for registration tend to be relatively small (i.e., come from an upper pyramid level), so this conversion takes only a few seconds, while converting a larger image/pyramid level to warp and save the full resolution may take over a minute (Supplementary Fig. 2a). However, while conversion may take some time, libvips uses "lazy evaluation", meaning that the WSI can then be warped and saved without having to load all of it into memory, making it ideal for large images such as WSI. Using this approach, VALIS is able to read, register, and save any slide that Bio-Formats or OpenSlide can open.

VALIS uses tissue features to find the transformation parameters, and therefore a lower resolution version of the image is used for feature detection and finding the displacement fields used in non-rigid registration. The lower resolution image is usually acquired by accessing an upper level of an image pyramid. However, if such a pyramid is unavailable, VALIS can use libvips to rescale the WSI to a smaller size. The default target shape is an image with a maximum dimension less than or equal to 850 pixels in width and/or height, without padding. We have found that increasing the size of the image used for registration does not always increase accuracy, and that gains tend to be small, despite the fact that WSI are frequently substantially larger than the default 850 pixels in width and/or height used by VALIS (Supplementary Fig. 2c). Figure 6a-b also provides examples of how registration using lower resolution images can translate to accurate alignment of the image at its native resolution.

### Preprocessing
For image registration to be successful, images need to look as similar as possible. In the case of IF, the DAPI channel is often the best option to use for registration. However, unless one is only working with H&E, a preprocessing method to make IHC images look similar must be used. The default method in VALIS is to standardize the color information from the image. This is accomplished by first converting the RGB image to the polar CAM16-UCS colorspace[39], setting $C = 0.2$ and $H = 0$ (other values can be used), and then converting back to RGB (RGB to CAM16-UCS conversion conducted using the color-science package for Python[40]). The transformed RGB image is then converted to greyscale and inverted, such that the background is dark, and the tissue bright. Benchmarking reveals that this preprocessing method produces registrations with relatively low mean error and small dispersion, indicating it yields more accurate and robust registration results than other frequently used methods, such as grayscale conversion, global histogram equalization of grayscale images, and contrast limited adaptive histogram equalization (CLAHE) of grayscale images (Supplementary Fig. 2b)[33].

After all images have been processed (IHC and/or IF), they are then normalized to one another to have similar distributions of pixel intensity values. The normalization method is inspired by[41], where first the 5th percentile, average, and 95th percentile of all pixel values is determined. These target values are then used as knots in cubic interpolation, and then the pixel values of each image are fit to the target value. A final denoising step, using total-variation (TV) denoising[42], is used to very mildly smooth the images while preserving edges, thereby reducing noise that can confound the registration.

### Mask creation
To help focus registration on the tissue, and thus avoid attempting to align background noise, VALIS generates tissue masks for each image. The underlying idea is to separate background (slide) from foreground

(tissue) by calculating how dissimilar each pixel's color is from the background color. The first step converts the image to the CAM16-UCS colorspace, yielding L (luminosity), A, and B channels. In the case of brightfield images, it is assumed that the background will be bright, and so the background color is the average LAB value of the pixels that have luminosities greater than 99% of all pixels. The difference to background color is then calculated as the Euclidean distance between each LAB color and the background LAB, yielding a new image, **D**, where larger values indicate how different in color each pixel is from the background. Otsu thresholding is then applied to **D**, and pixels greater than that threshold are considered foreground, yielding a binary mask. The final mask is created by using OpenCV to find, and then fill, all contours, yielding a mask that covers the tissue area. The mask can then be applied during feature detection and non-rigid registration to focus registration on the tissue.

## Rigid Registration

VALIS provides a groupwise rigid registration method for serially aligning any number of images, using feature detection and matching, with transformation matrices being estimated from the matched feature coordinates using the method described in[43], as implemented in scikit-image[44]. The default feature detector and descriptor are BRISK and VGG, respectively[45,46]. This combination was selected after conducting experiments wherein it was found that the BRISK/VGG pair consistently produced the largest number of good matches between four serially sliced H + E images in each of 27 samples (Supplementary Fig. 2d). Once features have been detected, all pairs of images are matched using brute force, with outliers removed using the RANSAC method[47].

RANSAC does an excellent job of removing most outliers, but some still get considered as inliers. Including the coordinates of such mismatched features will produce poor estimates of the transformation matrices that will align feature coordinates, resulting in inaccurate alignments. To address this, we perform a secondary match filtering using Tukey's box plot approach to outlier detection. Specifically, a preliminary rigid transformation matrix between source image $I_i$ and target image $I_j$, $M'_{i,j}$, is found and used to warp the source image's feature coordinates to their position in target image. Next, the Euclidean distances between warped source and target coordinates is calculated, $d'_{i,j}$. Inliers are considered to be those with distances between the lower and upper "outer fences", i.e., between Q1-3IQR and Q3 + 3IQR, where Q1 and Q3 are the first and third quartiles of $d'_{i,j}$, and IQR is the interquartile range of $d'_{i,j}$.

In order to increase the chances of successful registration, VALIS orders the images such that each image is surrounded by the two most similar images. This is accomplished by using matched features to construct a similarity matrix **S**, where the default similarity metric is simply the number of good matches between each pair of images. **S** is then standardized such the maximum similarity is 1, creating the matrix **S**′, which is used to create the distance matrix, $D = 1 - S'$. Hierarchical clustering is then performed on **D**, generating a dendrogram $T$. The order of images can then be inferred by optimally ordering the leaves of $T$, such that most similar images (i.e., leaves) are adjacent to one another in the series[48]. This approach was validated by reordering a shuffled list of 40 serially sliced H + E images from[12], where the original ordering of images is known. All 40 images were correctly ordered (Supplementary Fig. 2e), indicating that this approach is capable of sorting images such that each image is neighbored by similar looking images. While VALIS can sort images based on similarity, it is also possible to align the images in a specified order, based on the filename. This option is particularly useful if the aim is to construct a 3D image by registering serial sections, as shown in Fig. 6e.

Once the order of images has been determined, VALIS finds the transformation matrices that will serially rigidly warp each image to an adjacent image in the stack. Specifically, images are aligned towards

(not directly to) the image at the center of the series (or a reference image, if one is specified). For example, given N images, the center image is $I_{(\frac{N}{2})}$. Therefore, $I_{(\frac{N}{2})-1}$ is aligned to $I_{(\frac{N}{2})}$, then $I_{(\frac{N}{2})-2}$ is aligned to the rigidly warped version of $I_{(\frac{N}{2})-1}$, and so on. While the combination of RANSAC and Tukey's outlier detection methods remove most poor matches, VALIS performs a third match filtering step, which we refer to as neighbor match filtering. In this step, only features that are found in the image and its neighbors are considered inliers, the idea being that matches found in both neighbors reflect good tissue feature matches (Supplementary Fig. 3a). That is, the features used to align image $I_i$ and $I_{i-1}$ are the features that $I_i$ also has in common with $I_{i+1}$, and thus consequently that $I_{i-1}$ also has in common with $I_{i+1}$. This approach may be thought of as using a sliding window to filter out poor matches by using only features shared within an image's neighborhood. The coordinates of the filtered matches are then used to find the transformation matrix ($M_i$) that rigidly aligns $I_i$ to $I_{i-1}$ or $I_{i+1}$ (depending on the position in the stack).

After warping all images using their respective rigid transformation matrices, the group of images has been registered. However, one can optionally use an intensity-based method to improve the alignment between $I_i$ and its neighbor. One option is to maximize Mattes mutual information between the images, while also minimizing the distance between matched features[49]. Once optimization is complete, $M_i$ will be updated to be the matrix found in this optional step. This step is optional because the improvement (if any) may be marginal (distance between features being improved by fractions of a pixel), and it is time consuming.

## Non-Rigid Registration

Non-rigid registration involves finding 2D displacement fields, **X** and **Y**, that warp a "moving" image to align with a "fixed" image by optimizing a metric. As the displacement fields are non-uniform, they can warp the image such that local features align better than they would with a single global rigid transformation[50]. However, these methods require that the images provided are already somewhat aligned. Therefore, once VALIS has rigidly registered the images, they can be passed on to one to a non-rigid registration method.

Recall that rigid registration is performed on low resolution copies of the full image. However, it may be that the tissue to be aligned makes up only a small part of this image, and thus the tissue is at an overly low resolution (Fig. 2). However, the lack of detailed alignment can be overcome during the non-rigid registration step, which can be performed using higher resolution images. This is accomplished by creating a non-rigid registration mask, which is constructed by combining all rigidly aligned image masks, keeping only the areas where all masks overlap and/or where a mask touches at least one other mask. The bounding box of this non-rigid mask can then be used to slice out the tissue at a higher resolution from the original image (Fig. 2, additional examples in Supplementary Fig. 3c). These higher resolution images are then processed and normalized as before, warped with the rigid transformation parameters, and then used for non-rigid registration. This approach makes it possible to non-rigidly register the images at higher resolution without loading the entire high-resolution image into memory, increasing accuracy with low additional computational cost (due to re-processing the image).

Currently, VALIS can conduct non-rigid registration using one of three methods: Deep Flow, SimpleElastix, or Groupwise SimpleElastix[51–54]. In the case of the first two methods, images are aligned in the same order as used during rigid registration. Each image's displacement fields, $X_i$ and $Y_i$, are built through composition, allowing transformations to accumulate. A benefit of accumulating transformations serially is that distant features can be brought together, which may not occur if performing direct pairwise registration (Supplementary Fig. 3b). For the third method (Groupwise

SimpleElastix), this process of aligning pairs of images and composing displacement fields is not necessary, as it uses a 3D free-form B-spline deformation model to simultaneously register all of the images.

## Micro-registration

The above transformations are found using lower resolution images, but a second optional non-rigid registration can be performed using higher resolution images, which may improve alignment of "micro-features" not visible in the smaller images used for the initial registration. This is accomplished by first scaling and applying the above transformations to a larger image, and then performing a second non-rigid registration following the steps described above. This yields two new deformation fields, $\mathbf{X}_i'$ and $\mathbf{Y}_i'$, which can be added to the original displacement fields to get updated displacements that will align microstructures (Supplementary Fig. 3d). If the images are large (e.g., greater than 10 Gb), each image will be divided into tiles, which will be registered, and each tile's deformation fields stitched together to create the full size deformation fields. A caveat is that the increase in accuracy may be small and the computational cost high, so this step is optional (Supplementary Fig. 2c).

An alternative use of this micro-registration step can be to directly align images to a specified reference image after performing the default groupwise registration. This can be a good approach after using the groupwise method because all images should be closely aligned, and this step can improve the alignment directly to the reference image. An example of how this can improve registration to a target image is shown in Fig. 3a.

## Warping and Saving

Once the transformation parameters $\mathbf{M}_i$, $\mathbf{X}_i$, and $\mathbf{Y}_i$ have been found, they can be scaled and used to warp the full resolution image, which is accomplished using libvips. The warped full resolution image can then be saved as a pyramid ome.tiff image, with the ome-xml metadata being generated by the Python package *ome-types*, and image writing done using libvips[29,31]. Once saved as an ome.tiff, the registered images can be opened and analyzed using open-source software such as MCMICRO[26], QuPath[55], ImageJ[56] or commercially available software, such as Indica Labs HALO® (Albuquerque, NM, USA) or Visiopharm image analysis software. As the ome.tiff slides can be opened using libvips or Bio-Formats, one can also use the aligned slide in a more tailored analysis using custom code.

## Reporting summary

Further information on research design is available in the Nature Portfolio Reporting Summary linked to this article.

## Data availability

The ANHIR Grand Challenge dataset is available at https://anhir.grand-challenge.org/Data/. The ACROBAT Grand Challenge datasets are available at https://acrobat.grand-challenge.org/. The 3D datasets provided by[36] is available at http://urn.fi/urn:nbn:fi:csc-kata20170705131652639702. All other images are part of on-going studies and will be made available upon their publication. Source data are provided as a Source Data file[57].

## Code availability

Code is available on GitHub (https://github.com/MathOnco/VALIS)[27], PyPi (https://pypi.org/project/valis-wsi/), and DockerHub (https://hub.docker.com/r/cdgatenbee/valis-wsi). Full documentation can be found on Read the Docs (https://valis.readthedocs.io/en/latest/).

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

## Acknowledgements

The authors gratefully acknowledge funding by the National Cancer Institute via the Cancer Systems Biology Consortium (CSBC) U01CA232382, the Physical Sciences Oncology Network (PSON) U54CA193489 and support from the Moffitt Center of Excellence for Evolutionary Therapy. The authors also wish to acknowledge the role of the Breast Cancer Now Tissue Bank in collecting and making available the samples used in the generation of this publication, and the patients who donated to the Bank.

## Author contributions

C.D.G. developed the method, wrote the software, created the documentation, and performed the benchmarking and validation. A.M.B. designed, optimized, and performed experiments related to the CyCIF and brightfield colorectal cancer samples. R.J.C.S. designed, optimized, and performed experiments related to the cyclically stained IHC HNSCC samples. O.S. performed additional CyCIF registration benchmarking using other methods for comparison. S.P., G.M., and E.M. contributed to the writing of the manuscript. N.A., A.M., S.L., J.R.C.G., C.H.C., and T.A.G. graciously provided samples used to develop the method. M.R.T., T.A.G., and A.R.A.A. guided the research and supervised the writing of the manuscript. A.R.A.A. conceived and funded the study.

## Competing interests

The authors declare no competing interests.
