## [Peer Review File · Nature Communications]

nature portfolio

Peer Review FileReviewer comments

Reviewer #1 (Remarks to the Author):

The authors have developed a high throughput workflow for alignment of slide images across different modalities to facilitate spatial multiplexed studies. Certainly, this work is important for the field of digital pathology, especially making such registration technologies accessible to end users. As an example, using fine-grained image registration can tag cells given wash and re-staining of tissue, which is valuable to researchers and clinicians across many specializations. However, many existing wash and re-stain methods have been shown to introduce distortion artifacts. The authors have pointed out the utility of multiplexed immunofluorescence staining. I have a few comments on the scope of the comparisons:

- The authors allowed images to have a maximum size of 850 pixels. This is an unrealistic comparison and only focuses on macro-architectural target registration error. Certainly, micro-architectural features cannot be properly aligned on serial sections due to the depth-wise separation within the tissue block, but at minimum slides should be aligned at their native resolutions, 20x / 40x. Thus, the comparisons may not be convincing enough to encourage adoption of the platform. What macro-architectural features were compared to for the target registration error? If repeat measurements within case were considered, they would need to be adjusted for in statistical comparisons (e.g., mixed models).^[SEP] What are the RAM/GPU limitations of the platform?
- More than 4 datasets should be benchmarked for this journal.
- Elaboration on comparisons which yielded the chosen feature descriptor methods.
- Discussion on impact of stain deconvolution on registration accuracy.
- The greatest added value versus previous platforms would rapid and accurate cell-wise alignments on washed and re-stained sections, though I would like to see more information on how the number of washes impacts assessment, from registration to staining quality as potentially antibody binding may not be as strong after many rewashes. Highlighting the limitations of multiplexed IHC, characterized through this platform, could provide added value for the target audience, perhaps more so than existing models run on the multiplexing.
- I would recommend that the authors create a code ocean capsule to demonstrate the serial alignment across many stains at 20x/40x to illustrate the platform's utility.
- I did not find a limitations section in the paper, which is needed. At certain points in the manuscript, I had felt that the authors' were trying to "sell" the registration approach, e.g., "VALIS came to life" and the overuse of novel and robust. Such language should be removed from the manuscript.
- Species distribution models -> the authors may want to rename as spatial autocorrelation model, should describe this assessment in the methods section or consider applying the SPARK methodology or something similar. The authors should apply other methods to the CyCIF dataset and discuss how downstream modeling is impacted with improper registration.
- In terms of journal scope, this article dives a bit too deeply into the methods, which can make it difficult to digest by the target audience and its broad utility may be under appreciated. Most of the methods could be moved to supplementary and instead further details of the experimental design and significance of each dataset could be added instead.

In sum, this is a nice informatics paper with potential utility for informatics and basic sciences researchers looking to develop and incorporate spatial multiplexing to inform cancer biology research. The added value of this work is introducing software that is both scalable and accessible. However, the novelty of this work is limited as I have seen similar renditions of the proposed methodology (whether or not such methods were added to official software distributions) and does not present new results. Neither does it offer practical advice (maybe in supplementary materials) of how users can get started and what machinery is required (e.g., compute cluster, local machine, GPU, etc.). While this is fine when presenting new software, the article may be better targeted towards a pathology or medical informatics journal as study significance is unclear.

Reviewer #2 (Remarks to the Author):

Title: VALIS: Virtual Alignment of pathology Image Series

This paper presented the Virtual Alignment of pathology Image Series (VALIS) software for the registration of whole slide images (WSI) that were serially sliced and/or cyclically stained. The VALIS approach consists of two stages. In the first stage, rigid transformations were estimated to warp each image to its previous image in the series. In the second stage, rigidly aligned images were registered by using one of the following three approaches: Deep Flow, SimpleElastix, or Groupwise SimpleElastix. The VALIS approach was evaluated on multiple datasets with various levels of difficulty to register.

Major limitation: The methodological novelty of this paper is limited. The proposed rigid registration approach is similar to prior approaches except for the preprocessing steps and the removal of feature outliers. Deformable registrations were performed using prior approaches.

Detailed comments:

1. Image Preprocessing. Why did the authors resize all images to the same dimension in height and width? This would change the physical size of the tissues and may make the registration problem harder.
2. The authors have developed an algorithm to estimate the order of images if it is unknown. However, no results have been provided regarding the accuracy of this algorithm.
3. Details of the registration approaches are missing, which affect the reproducibility of the work. For example, how did the authors handle large rotations in WSI (for example in Fig. 3c)? Also, what is the stopping criterion? What is the similarity loss function?
4. Registration error was evaluated by using features that were used by the VALIS software, which is not a fair comparison. I would suggest the authors randomly select evenly distributed landmarks for the evaluation.
5. It would be interesting to see how VALIS can be generalized to other datasets. For example, whole-mount histopathology images of the brain [A], and the prostate [B], etc.
A. Mancini, Matteo, et al. "A multimodal computational pipeline for 3D histology of the human brain." *Scientific reports* 10.1 (2020): 1-21.
B. Rusu, Mirabela, et al. "Registration of presurgical MRI and histopathology images from radical prostatectomy via RAPSODI." *Medical physics* 47.9 (2020): 4177-4188.
6. Figure 8. VALIS may also be used to facilitate the registration WSI with other imaging modalities [A, B], e.g., MRI.
7. What does the term "rotation invariant" mean in this paper? Please clarify.

Reviewer #3 (Remarks to the Author):

The paper describes a software-toolbox that is composed of a combination of new (tissue pre-processing and automatic sorting) and previously existing (such as RANSAC, simple elastix, and deepflow) registration building blocks. Next to the combination of the different building blocks, the contribution of the paper seems to focus on the method to automatically sort the order of a stack of serial sections and a RANSAC-based rigid registration. The paper could be improved by making clearer which parts are implementations of previous work and where novel contributions have been made.

The authors compare their toolbox against other methods on a currently private dataset. However, the comparison seems to overlook a few methods that have been shown to be successful in the ANHIR challenge (Borovec et al., 2020, doi: 10.1109/TMI.2020.2986331) and which follow very similar concepts (i.e. the method labeled AGH seems to follow a very similar strategy based on RANSAC + existing deformable registration).

Also, evaluating the method as part of the ANHIR challenge (which is still open for submissions) would provide an impartial evaluation that would not suffer the potential bias of re-using the detected features that compute the registration for evaluation.

Since VALIS is partially a collection of existing tools, it might be worthwhile to compare the individual building blocks of VALIS against their respective state of the art instead of comparing multiple systems as black boxes.

- How well do the other methods perform if images are preprocessed as proposed?
- How does the RANSAC-based approach compare against other rigid registration methods on low-resolution images?

In a final application chapter, the authors apply the VALIS toolbox to perform an analysis with multiple markers in the tumor microenvironment. They demonstrate that the software is fit to interdisciplinarily answer a research question. However, this section addresses a question that goes far beyond the alignment of the slides discussed in the first part and already is a small paper in itself. It might be of more value in a separate publication.

I made a few additional comments directly in the manuscript (and apologize if my handwriting might not be legible).

Reviewer #1 (Remarks to the Author):

The authors have developed a high throughput workflow for alignment of slide images across different modalities to facilitate spatial multiplexed studies. Certainly, this work is important for the field of digital pathology, especially making such registration technologies accessible to end users. As an example, using fine-grained image registration can tag cells given wash and re-staining of tissue, which is valuable to researchers and clinicians across many specializations. However, many existing wash and re-stain methods have been shown to introduce distortion artifacts. The authors have pointed out the utility of multiplexed immunofluorescence staining. I have a few comments on the scope of the comparisons:

- The authors allowed images to have a maximum size of 850 pixels. This is an unrealistic comparison and only focuses on macro-architectural target registration error. Certainly, micro-architectural features cannot be properly aligned on serial sections due to the depth-wise separation within the tissue block, but at minimum slides should be aligned at their native resolutions, 20x / 40x. Thus, the comparisons may not be convincing enough to encourage adoption of the platform. What macro-architectural features were compared to for the target registration error? If repeat measurements within case were considered, they would need to be adjusted for in statistical comparisons (e.g., mixed models). - What are the RAM/GPU limitations of the platform?

We agree that being able to register the image at its native resolution would be ideal, and have added two new steps to increase the resolution of the images used for registration. The first new step is performing non-rigid registration on a higher resolution version of the image used for the rigid registration (Figure 3 below, lines 328-359 in main text). The computational cost of this is reduced by “slicing” out only the tissue from the higher resolution image, making it possible to perform registration at higher resolution without having to load the full image into memory. The second step, which we refer to as micro-registration, performs a secondary non-rigid alignment after the initial groupwise registration, using even larger images, potentially even the full size image (Figures 3 and 4g below, lines 361-377 in main text).

We tested the updated method in two Grand Challenges, the 2019 Automatic Non-rigid Histological Image Registration (ANHIR) grand challenge (lines 427-501 in main text), and the 2022 Automatic Registration of Breast Cancer Tissue (ACROBAT) grand challenge, each of which provides hand annotated landmarks that have been matched across image pairs. We conducted experiments wherein we performed the micro-registration step with various degrees of downsampling: the default (850 pixels), 25%, 50%, or 75% of the full resolution image, or no down sampling at all (i.e. 100% resolution). To our surprise, we found that performing this higher resolution registration did not

provide a significant improvement in accuracy (Figure 4a below). This can also be visualized in Figure 4b, which shows regions of interest in two full resolution registered images (>50K pixels in width and height) whose registration parameters were found on much smaller (1000 pixels) images. We believe this is because registration of the small images also aligns major tissue features, which “bring along” the micro-architectural features during their movement to the registered coordinate space. As the gains of micro-registration may be small, we decided to offer it as an optional step to refine the alignment.

Regarding RAM/GPU limits, VALIS uses libvips to work with the full resolution WSI. This is ideal, because libvips does not have to load the image into memory, and so it is capable of warping and saving very large images. We haven’t been able to find an official limit documented by libvips, but the author (John Cuppit, user “jcuppit”) claims “There should be (almost) no limit. I regularly process 300,000 x 300,000 pixel slide images” (<https://github.com/libvips/libvips/issues/1712>). In practice we have registered and saved WSI that exceed 38Gb (uncompressed), and have heard from users who have done the same with images having dimensions up to 107,074 x 91,313 pixels (width, height). Currently, VALIS is not setup to run on GPU, but it is certainly something we will look into.

- More than 4 datasets should be benchmarked for this journal.

We agree, and have replaced those four datasets with the publicly available ANHIR and ACROBAT Grand Challenge datasets mentioned previously (Figure 1 below, lines 427-501 in main text). In the ACROBAT Grand Challenge, VALIS has ranked first among the methods with available code, and second overall. It is also currently ranked in first place on the ACROBAT Grand Challenge’s validation leaderboard, which remains open for submissions. Among the publicly available methods, VALIS is a close second in the ANHIR Grand Challenge, being behind the next highest ranked published method (SFG) by only a few pixels (Figure 4h below). While we have included the ANHIR results in the updated manuscript, we unfortunately cannot include the ACROBAT Grand Challenge results until the paper describing the challenge has been published.

- Elaboration on comparisons which yielded the chosen feature descriptor methods.

We have now described the experiment we conducted to select the default feature detector and descriptors (Figure 4c below, lines 276-279 in main text). Briefly, we had 27 samples that had four serially sliced H&E images, and we assumed that a good feature detector/descriptor pair would be one that can detect a large number of “good” matches between each pair of serial H&E slices. Using this logic, we determined the average number of “good” matches (i.e. those considered as inliers after RANSAC outlier removal) between each pair of H&E images (N=6 pairs) for all 27 samples. By far, the BRISK/VGG pair yielded the largest number of good matches between each pair of H&E images.

- Discussion on impact of stain deconvolution on registration accuracy.

An advantage VALIS has over some other WSI registration methods is that it does not rely on stain deconvolution. The preprocessing method is stain agnostic, and instead of enhancing a common stain through color deconvolution, our method standardizes “colorfulness”, followed by intensity normalization. As discussed in more detail below, we acknowledge that repeated stain/wash cycles would have an impact on stain quality, and thus stain deconvolution, but registration accuracy does not appear to be affected.

- The greatest added value versus previous platforms would rapid and accurate cell-wise alignments on washed and re-stained sections, though I would like to see more information on how the number of washes impacts assessment, from registration to staining quality as potentially antibody binding may not be as strong after many rewashes. Highlighting the limitations of multiplexed IHC, characterized through this platform, could provide added value for the target audience, perhaps more so than existing models run on the multiplexing.

We have now noted in the Discussion section that there are limits to how many stain/wash cycles can be performed, which in turn limits the number of markers that can be merged into a single image. In particular, we mention that the authors of t-CyCIF state that tissues can reliably undergo 8-10 rounds of stain/wash, and possibly up to 20 in some cases (Lin et al., 2018). We also provide some practical advice on how to maximize the number of markers used in the cyclic staining. However, it is not our intention to tether VALIS to a particular staining protocol, and thus it seems that assessing staining quality is outside the scope of this work. We believe assessment is best left to those performing the staining, as suggested in the t-CyCIF paper (Lin et al., 2018). We also suspect that quality will improve with time as technological advances are made.

Regarding registration accuracy, the cyclically stained IHC samples we tested underwent between 13-20 stain/wash cycles, and all registered successfully (Figure 2 below). In fact, each of those complete stain/wash cycles was performed on one of four serial slices, all of which (N=69) were successfully registered as essentially one sample, potentially yielding a 69-plex image derived from IHC. This was repeated for 20 samples, and in all cases registration was successful. Similarly, we did not experience issues registering our 6 CyCIF samples, each of which underwent 11 rounds of staining (Figure 2 below). We would argue that, at least with that stain/wash protocol, the number of stain/wash cycles did not have a significant impact on registration quality.

- I would recommend that the authors create a code ocean capsule to demonstrate the serial alignment across many stains at 20x/40x to illustrate the platform’s utility.

This is an excellent suggestion, especially if the goal is to provide an example of aligning images using the full resolution 20x/40x image. It is something we will look into. However, we would like to point out that all code is available on GitHub (<https://github.com/MathOnco/valis>) and PyPi (<https://pypi.org/project/valis-wsi/>), with full documentation on ReadTheDocs (<https://valis.readthedocs.io/en/latest/>). We also provide several examples of how to use VALIS, as well as a two example datasets to

illustrate those examples. These include lower resolution versions of the original slides, but the process is the same, and we have used VALIS to read, register, and save those original 40x images. Regarding access or installation, we have yet to receive any issues related to those topics. To the contrary, we have been told it is easy to use and install. Even so, we are working to make it even easier by putting VALIS on conda-forge, which will remove the need to install libvips and/or Openslide separately.

- I did not find a limitations section in the paper, which is needed. At certain points in the manuscript, I had felt that the authors' were trying to "sell" the registration approach, e.g., "VALIS came to life" and the overuse of novel and robust. Such language should be removed from the manuscript.

While we have not created a dedicated limitations section, we have discussed them throughout out the paper. We have not run into memory issues, and have been able to register 69 images simultaneously, so we have discussed limitations in the number of staining rounds, and that VALIS' non-rigid registration approach may not be ideal for 3D reconstruction.

In trying to illustrate how image registration can be used in lieu of highly multiplexed staining, we can see how it comes across as us trying to "sell" the registration approach. Our aim is more to show that it can be a viable alternative to those without access to technology/expertise that allows them to perform very high multiplexing on a single image. We also want to illustrate how registration can help enable spatial analyses using existing datasets. We have reduced the use of "novel" and "robust", and have removed the text describing how "VALIS came to life". In its place we summarize VALIS' unique contributions (lines 390-423 in main text).

- Species distribution models -> the authors may want to rename as spatial autocorrelation model, should describe this assessment in the methods section or consider applying the SPARK methodology or something similar. The authors should apply other methods to the CyCIF dataset and discuss how downstream modeling is impacted with improper registration.

Species distributions models (SDM) come from the field of ecology, where they are used determine which environmental factors create hospitable environments for the species under study. The tool we used to fit the SDM, *dismo* (Hijmans, Phillips, Leathwick, & Elit, 2017), was developed by and for landscape ecologists, and so we feel it would be best to leave the model's description as a species distribution model in place. Regarding additional analyses, our primary aim to is provide examples of spatial analyses when registration accuracy is high enough to perform cell segmentation (CyCIF), and when it is not quite there (IHC). Other reviewers have suggested removing the more detailed analyses as it is somewhat tangential to the method, but we think keeping it in place will be of value to those who are more interested in the application than the method itself. Keeping the analyses to these two minimal examples may be a good middle ground.

- In terms of journal scope, this article dives a bit too deeply into the methods, which can make it difficult to digest by the target audience and its broad utility may be underappreciated. Most of the methods could be moved to supplementary and instead further details of the experimental design and significance of each dataset could be added instead.

While we think the method is interesting, we too feel that the more exciting and useful aspect of VALIS is the ability to easily register and save WSI in their native resolution with state-of-the-art accuracy. As such, we can see how the methods may be better placed in the supplemental section, given the scope of this journal. However, other reviewers have focused only on the method and expressed limited interest in the application, which leads us to conclude that there is also equal interest in how exactly VALIS performs the registration. For this reason, we think it may be best to leave the methods in the main document.

In sum, this is a nice informatics paper with potential utility for informatics and basic sciences researchers looking to develop and incorporate spatial multiplexing to inform cancer biology research. The added value of this work is introducing software that is both scalable and accessible. However, the novelty of this work is limited as I have seen similar renditions of the proposed methodology (whether or not such methods were added to official software distributions) and does not present new results. Neither does it offer practical advice (maybe in supplementary materials) of how users can get started and what machinery is required (e.g., compute cluster, local machine, GPU, etc.). While this is fine when presenting new software, the article may be better targeted towards a pathology or medical informatics journal as study significance is unclear.

We are happy to hear the reviewer believes this work will be of use to those who would like to incorporate spatial multiplexing into their research. As that is our primary goal, we think we may have not spent enough time emphasizing what makes VALIS unique. Regarding novelty, it is true that VALIS uses existing feature detectors, feature descriptors, and non-rigid registration methods potentially used in other approaches, but it does so in a new way. While feature matches are used in typical way to find rigid registration parameters later in the pipeline, they are also used early on to order the images by similarity, which in turn increases the registration accuracy and has several benefits:

1. Sorting the images by similarity and then aligning serially ensures that images are aligned to the most similar looking image, increasing the chances of successful registration for each image pair.
2. Feature match quality is improved by using only the matches found in both neighbors, which should be indicative of strong tissue features. This reduces the number of poor matches included in the estimation of rigid registration parameters, thereby increasing registration accuracy. This is only possible because the images have been sorted by similarity.
3. Ordering the images and aligning them serially solves the non-trivial problem of selecting a reference image. If aligning more than two images, selecting the wrong

reference image can cause a registration to fail. This can be especially challenging if an H&E image is not available, as it may not be obvious which image looks most similar to the rest.

4. Because transformations accumulate, distant features in dissimilar images can be better aligned than might be possible if only performing pairwise registration. This too is only possible because the images have been sorted and aligned serially (Figure 4f below).
5. Since images are sorted and aligned serially, any number of images can be registered at the same time, in contrast to conducting several pairwise alignments. Again, this is only possible because we use feature matches to order images by similarity.
6. The ability to rapidly align a large number of images makes VALIS a nice option when the goal is to construct a multiplex image from several rounds of staining, such as with CyCIF.

We hope that this better illustrates how VALIS' approach is unique, despite using some existing registration building blocks. We would also argue that similar methods that are not published, either as a paper or software, are of interest but limited use. It's no trivial task to re-implement a method, especially because "the devil is in the details", which may or may not be well described.

Another unique feature of VALIS, which we have now further emphasized, is that VALIS is able to read, register, and save massive multi-gigapixel WSI in their native resolution. In contrast, most other methods only read a handful of formats, which sometimes doesn't include formats unique to WSI, such SVS, MRXS, OME-TIFF, etc... Of the published methods we have found, none are able save the images in pyramid ome.tiff format, which is capable of storing these very large WSI. We believe this allows VALIS to be used in more real-world cases than other existing methods.

Regarding practical advice, we have provided full documentation on ReadTheDocs (<https://valis.readthedocs.io/en/latest/>), along with examples and installation instructions in the README, which is also available on ReadTheDocs, GitHub (<https://github.com/MathOnco/valis>), and PyPi (<https://pypi.org/project/valis-wsi/>). Issues and practical advice can be addressed using GitHub's Issues tracker (<https://github.com/MathOnco/valis/issues>). We think these venues are better suited for practical advice than a supplement, as they can be updated as features are added.

Reviewer #2 (Remarks to the Author):

Title: VALIS: Virtual Alignment of pathology Image Series

This paper presented the Virtual Alignment of pathoLogY Image Series (VALIS) software for the registration of whole slide images (WSI) that were serially sliced and/or cyclically stained. The VALIS approach consists of two stages. In the first stage, rigid transformations were estimated to warp each image to its previous image in the series.

In the second stage, rigidly aligned images were registered by using one of the following three approaches: Deep Flow, SimpleElastix, or Groupwise SimpleElastix. The VALIS approach was evaluated on multiple datasets with various levels of difficulty to register.

Major limitation: The methodological novelty of this paper is limited. The proposed rigid registration approach is similar to prior approaches except for the preprocessing steps and the removal of feature outliers. Deformable registrations were performed using prior approaches.

It is true that VALIS uses existing feature detectors, feature descriptors, and non-rigid registration methods, but it does so in a new way (Figure 3 below). While feature matches are used in typical way to find rigid registration parameters later in the pipeline, they are also used early on to order the images by similarity, which in turn increases the registration accuracy and has several benefits:

1. Sorting the images and then aligning serially ensures that images are aligned to the most similar looking image, increasing the chances of successful registration for each image pair.
2. Feature match quality is improved by using only the matches found in both neighbors, which should be indicative of strong tissue features. This reduces the number of poor matches included in the estimation of rigid registration parameters, thereby increasing registration accuracy. This is only possible because the images have been sorted by similarity.
3. Ordering the images and aligning them serially solves the non-trivial problem of selecting a reference image. If aligning more than two images, selecting the wrong reference image can cause a registration to fail. This can be especially challenging if an H&E image is not available, as it's not obvious which image looks most similar to the rest.
4. Because transformations accumulate, distant features in dissimilar images can be better aligned than might be possible if only performing pairwise registration. This too is only possible because the images have been sorted and aligned serially.
5. Since images are sorted and aligned serially, any number of images can be registered at the same time, in contrast to conducting several pairwise alignments. Again, this is only possible because we use feature matches to order images by similarity. This feature also makes VALIS uniquely suited to registering large rounds of cyclically stained images, such as CyCIF.

In addition to these unique characteristics arising from ordering the images, we have updated VALIS to with two new steps to increase registration accuracy. The first update comes in the form of performing non-rigid registration on higher resolution images than those used in rigid registration (Figure 3 below, lines 336-348 in main text). This step involves finding the tissue after rigid registration, extracting it from the higher resolution image, and then re-preprocessing for non-rigid registration. This, in turn, helps align the finer details in the image.

The second addition involves refining the registration in a step we call micro-registration, where the goal is to align the “micro-architectural” features in the image (lines 360-377 in main text). This step performs a second non-rigid registration on a very high resolution copy of the image, potentially at native resolution. However, we found that the gains in accuracy are limited, and the trade off in computation time may not always be “worth it” (Figure 4a below). As such, we have offered this functionality as an optional step after the primary registration.

We hope that the above better illustrates how VALIS’ approach is unique, despite using some common image registration building blocks.

While appropriate selection of feature detectors and non-rigid methods is crucial, we would argue that the importance of pre-processing should not be underestimated. No matter how good a feature detector/descriptor or non-rigid method is, they all assume that images will look similar. The new pre-processing methods we describe do that job well and are stain agnostic, meaning they do not require stain deconvolution (which requires providing or estimating a stain matrix, a unique challenge in itself). This generalizability is what helps VALIS perform well across a wide variety of datasets.

Likewise, outlier removal is also critically important. Again, it is likely that matched features using even the “best” feature detector/descriptor will include outliers, which would result in poor alignments. Outlier removal is so crucial that we have even added a third step to remove outliers, based on Tukey’s outlier detection method (lines 281-291 in main text). Therefore, there are now three steps for outlier removal: RANSAC (traditional approach) and Tukey during initial matching, and neighbor match filtering during serial rigid registration. Combined, these steps go a long way in ensuring that only “good” matches are used to find rigid registration parameters, in turn producing better rigid registrations. Good rigid-registration in turn increases the quality of non-rigid registrations, as those methods typically require the images already be closely aligned.

Detailed comments:

1. Image Preprocessing. Why did the authors resize all images to the same dimension in height and width? This would change the physical size of the tissues and may make the registration problem harder.

We should clarify that images were resized so that their maximum dimensions were the same, not necessarily that they were resized to have the same shapes. As such, all images maintained their original aspect ratio. Performing registration on smaller images is a common approach, as it actually makes registration easier. This is because the lower resolution images (which have the same aspect ratio as the original) essentially contain less noise/details that could get mismatched, leaving the major tissue landmarks as the most prominent features. Aligning those tissue level landmarks can translate into aligning micro-features present in the full size image but not in the small image. This is something we have shown in new experiments in the current manuscript (Figure 4b below). Another benefit of performing the registration on smaller images is that it is much faster than processing, detecting features, and non-rigidly aligning a larger image. Again, we show

the tradeoff between accuracy and computation-time in our new experiments (Figure 4a below). We should note two things though. First, even though registration is performed on the smaller image, the transformations can be scaled up for use on the larger image, meaning that the registered WSI can be saved in the native resolution using the registration parameters found using the lower resolution images. Second, we now provide an option to perform a second non-rigid registration on larger images, so as to improve alignment of micro-features (Figures 3 and 4g below).

2. The authors have developed an algorithm to estimate the order of images if it is unknown. However, no results have been provided regarding the accuracy of this algorithm.

This is a good point. We have now provided a description of how we validated the accuracy of this order approach (Figure 4d below, lines 300-303 in main text). Briefly, we shuffled the order of 40 serially sliced images whose order is known, and then applied this sorting method to the randomly ordered images. This approach correctly ordered all 40 images, indicating that it is able accurately order images such that each image is surrounded by its most similar looking images.

3. Details of the registration approaches are missing, which affect the reproducibility of the work. For example, how did the authors handle large rotations in WSI (for example in Fig. 3c)? Also, what is the stopping criterion? What is the similarity loss function?

Thanks for highlighting this. Regarding handling large rotations, we estimated the transform given matching feature coordinates. This was accomplished using scikit-image, which uses the method described in (Umeyama, 1991). We have added this detail to the manuscript. To ensure reproducibility, we have shared all code on GitHub and made the package available for installation using PyPi.

4. Registration error was evaluated by using features that were used by the VALIS software, which is not a fair comparison. I would suggest the authors randomly select evenly distributed landmarks for the evaluation.

We agree this was not the ideal approach to use when comparing registration methods, and have replaced the private benchmarking dataset with two Grand Challenges, the 2019 Automatic Non-rigid Histological Image Registration (ANHIR) grand challenge (lines 426-501 in main text), and the 2022 Automatic Registration of Breast Cancer Tissue (ACROBAT) grand challenge, each of which provides hand annotated landmarks that have been matched across image pairs. In the ACROBAT grand challenge, VALIS has ranked first among the methods with available code, and second place overall. It is also currently ranked in first on the ACROBAT validation leaderboard. Of the publicly available methods on the ANHIR leaderboard, VALIS placed a close second, with its accuracy differing from first place published method, SFG, by a few pixels (Figure 4h below).

Unfortunately, we are unable to include the ACROBAT results in this paper as the paper describing the grand challenge must be published first.

5. It would be interesting to see how VALIS can be generalized to other datasets. For example, whole-mount histopathology images of the brain [A], and the prostate [B], etc.

A. Mancini, Matteo, et al. "A multimodal computational pipeline for 3D histology of the human brain." *Scientific reports* 10.1 (2020): 1-21.

B. Rusu, Mirabela, et al. "Registration of presurgical MRI and histopathology images from radical prostatectomy via RAPSODI." *Medical physics* 47.9 (2020): 4177-4188.

In addition to the ACROBAT and ANHIR benchmarking datasets mentioned above, we have also tested VALIS with an additional 8 datasets that came from tissues of various origins and modalities (Figure 2 below). This internal dataset contained a total of 613 samples, each having between 2-69 images. Combining the ANHIR and internal datasets allowed us determine how VALIS generalizes by performing registration on a total of 13 tissue types (N=4,887 images): colon adenocarcinomas, colon carcinomas, human breast, lung lobes, lung lesions, human kidney, mice kidney, mammary glands, gastric mucosa, gastric adenocarcinoma, squamous cell carcinoma of the head and neck (HNSCC), ductal carcinoma in situ (DCIS), and tumor microarray cores (TMA) from human ovarian cancer. VALIS performed well on most images, ranking as the most accurate publicly available method in the ANHIR Grand Challenge. Error in the internal dataset was generally low, in some cases with accuracy being high enough for cell segmentation (Figure 4b below, and Figure 7 in the main manuscript).

6. Figure 8. VALIS may also be used to facilitate the registration WSI with other imaging modalities [A, B], e.g., MRI.

While we have tested VALIS with brightfield and immunofluorescence images, we have not yet tested it with MRI. While it would be interesting to do so, our current primary focus is on histology, as MRI registration seems less of an open problem. Even so, testing on MRI would certainly be worthwhile and is something we would like to explore in the future.

7. What does the term "rotation invariant" mean in this paper? Please clarify.

Rotation invariant refers to the ability to find rotations that can help align images. Such rotations are not estimated using methods like phase cross correlation used in ASHLAR (Muhlich, Chen, Russell, & Sorger, 2021) or the method described in ReStain (Jiang, Larson, Prodduturi, Flotte, & Hart, 2019; Muhlich et al., 2021), which limit transformations to shifting the images up/down, left/right. This, in turn, means that methods that are not rotation invariant likely need to have the images be already closely aligned. In contrast, the similarity transform estimated by VALIS does find which rotations are needed to align images, and so can better align WSI that were not placed on the slide consistently (sometimes upside-down).

Reviewer #3 (Remarks to the Author):

The paper describes a software-toolbox that is composed of a combination of new (tissue pre-processing and automatic sorting) and previously existing (such as RANSAC, simple elastix, and deepflow) registration building blocks. Next to the combination of the different building blocks, the contribution of the paper seems to focus on the method to automatically sort the order of a stack of serial sections and a RANSAC-based rigid registration. The paper could be improved by making clearer which parts are implementations of previous work and where novel contributions have been made.

We agree with the reviewer, and have tried to better describe VALIS' novel contributions. We believe VALIS makes two important contributions: 1) a new groupwise registration method, 2) the ability read, register, and save huge multi-gigapixel WSI. Regarding the method itself, it is certainly true that the feature detectors, feature descriptors, and non-rigid registration methods (e.g. the building blocks) are not new. However, the way in which we use their products (i.e. matched feature coordinates and deformation fields) is new (Figure 3 below). This originates with using all pairwise matches to create and sort a distance matrix, which orders the images based on similarity. Using this ordering, the images can be aligned serially, which enables several unique approaches that increase registration accuracy:

1. Sorting the images and then aligning serially ensures that images are aligned to the most similar looking image, increasing the chances of successful registration for each image pair.
2. Feature match quality is improved by using only the matches found in both neighbors, which should be indicative of strong tissue features. This reduces the number of poor matches included in the estimation of rigid registration parameters, thereby increasing registration accuracy. This is only possible because the images have been sorted by similarity.
3. Ordering the images and aligning them serially solves the non-trivial problem of selecting a reference image. If aligning more than two images, selecting the wrong reference image can cause a registration to fail. This can be especially challenging if an H&E image is not available, as it's not obvious which image looks most similar to the rest.
4. Because transformations accumulate, distant features in dissimilar images can be better aligned than might be possible if only performing pairwise registration. This too is only possible because the images have been sorted and aligned serially.
5. Since images are sorted and aligned serially, any number of images can be registered at the same time, in contrast to conducting several pairwise alignments. Again, this is only possible because we use feature matches to order images by similarity. This feature makes VALIS a good option when the goal is to construct a highly multiplexed image from several rounds of staining, such as CyCIF.

In addition to these unique characteristics arising from ordering the images, we have updated VALIS to with two new steps to increase registration accuracy. The first update comes in the form of performing non-rigid registration on higher resolution images than those used in rigid registration (Figure 3 below, lines 336-348 in main text). This step involves finding the tissue after rigid registration, extracting it from the higher resolution image, and then re-preprocessing for non-rigid registration. This, in turn, helps align the finer details in the image.

The second addition involves refining the registration in a step we call micro-registration, where the goal is to align the “micro-architectural” features in the image (lines 361-377 in main text). This step performs a second non-rigid registration on a very high resolution copy of the image, potentially at native resolution. However, we found that the gains in accuracy are limited, and the trade off in computation time may not always be “worth it” (Figure 4a below). As such, we have offered this functionality as an optional step after the primary registration.

We hope the above better illustrates the novelty of VALIS’ approach, which provides a new multi-resolution groupwise registration method, despite using some existing registration building blocks.

The second major contribution of VALIS provides is the ability to read, register, and save the huge multi-gigapixel WSI at their native resolution. Most other WSI registration methods can open a limited number of formats, and are unable to save the full resolution WSI. VALIS solves this problem by using Bio-formats and OpenSlide to convert slides to libvips Image objects, which can then be used warp and save registered images as pyramid ome.tiff files with their original metadata. The libvips package is ideal for working with large images, as does not require loading the full image into memory. Again, libvips is a pre-existing package, but we have provided the bridge between Bio-formats and libvips (which already supports OpenSlide) that makes it possible to read, register, and save WSI. This is particularly important in the case of immunofluorescence images, as OpenSlide only supports RGB images. A bonus of providing this bridge is that the use of VALIS is not limited to image registration, as researchers can use it to read WSI as libvips Images, which can readily perform operations on large images.

Within the manuscript, we have tried to emphasize these two contributions by describing registration of WSI as a two part problem: the first is the registration itself, the second is actually applying it to huge multi-gigapixel images that cannot be loaded into memory. Without this latter feature, we would argue that use of the registration method is limited, as WSI images are often saved in special formats and huge in size.

The authors compare their toolbox against other methods on a currently private dataset. However, the comparison seems to overlook a few methods that have been shown to be successful in the ANHIR challenge (Borovec et al., 2020, doi: 10.1109/TMI.2020.2986331) and which follow very similar concepts (i.e. the method labeled AGH seems to follow a very similar strategy based on RANSAC + existing deformable registration).

Also, evaluating the method as part of the ANHIR challenge (which is still open for submissions) would provide an impartial evaluation that would not suffer the potential bias of re-using the detected features that compute the registration for evaluation.

We would like to thank the reviewer for pointing us to the ANHIR challenge dataset. We agree that it provides a more unbiased assessment of VALIS's performance. We have benchmarked VALIS using this dataset by conducting several of experiments and have found its median median rTRE score to be between 0.00192. and 0.00172 (Figure 1 below, lines 426-501 in main text). The "worse" score of 0.00192 would tie it with HistoReg, for which we have found the publication and provide a comparison to in the manuscript. A major benefit VALIS has over HistoReg is related to VALIS' second contribution, i.e. the ability to read, register, and save very large images saved in a wide variety of formats. HistoReg is limited to reading a few formats (none of which are WSI specific formats, e.g. SVS, MRXS, OME-TIFF, etc...), and saves registered images in the NIfTI-1 format via the Insight Toolkit (ITK), which has a size limit of 32,767 pixels in width or height (Larobina & Murino, 2014; McCormick, Liu, Ibanez, Jomier, & Marion, 2014), meaning that HistoReg cannot read or write larger WSI saved in specialized formats, a common use case. Furthermore, HistoReg is limited to brightfield images, while VALIS can do that and also register and merge immunofluorescence images. These things combined allow VALIS to be used under a wider variety of use cases.

VALIS' best score of 0.00172 places it closely behind the best published method, SFG, but ahead of the mentioned AGH method. The SFG method has two entries on the leaderboard, with the first being 0.00067, and the latter being 0.00164. However, based on the paper, it appears the first (better) score was calculated using a private set landmarks created by their own expert (appendix Table XI in the SFG paper), while the second score is based on the ANHIR landmarks all other contributors had access to (main Table III in the SFG paper). Therefore, if we would argue that only the second value of 0.00164 should be used to compare entries. As shown in Figure 4h below, the difference between 0.00164 (SFG 1st place) and 0.00172 (VALIS 2nd place) actually only reflects a improvement of a few pixels. As in the comparison with HistoReg, VALIS' other key advantage over SFG is that it can actually read, warp, and save the WSI at native resolution. In fact, SFG "only" takes the image and returns the transformations, leaving it up to the user to figure out how to read the slide, apply it to the full resolution WSI, and then save the results. This is not trivial, as working with WSI can be challenging to work with simply due to their sheer size, as they often do not fit into memory. This is not intended to be a "knock" against SFG, because its sole aim is find the non-rigid transformations, but we do hope it illustrates that VALIS is more accessible while still having state of the art accuracy.

In addition to the ANHIR Grand Challenge, we have also benchmarked VALIS using the Automatic Registration of Breast Cancer Tissue (ACROBAT) grand challenge. In the ACROBAT Grand Challenge, VALIS has ranked first among methods with available code, and second overall. It is also currently ranked first in the validation leaderboard.

Unfortunately, we cannot include these results in this paper, as the paper describing the grand challenge has yet to be published.

We would also like to point out that we tested VALIS on 613 other datasets that came from a variety of tissues, staining protocols, resolutions, and imaging modalities (brightfield or immunofluorescence). Even with this diverse “real world” dataset VALIS produced accurate alignments in most cases (Figure 2 below). Therefore, in this paper we have tested VALIS using 5,138 image pairs, with most registrations being successful. We believe this is a testament to VALIS’ generalizability.

Since VALIS is partially a collection of existing tools, it might be worthwhile to compare the individual building blocks of VALIS against their respective state of the art instead of comparing multiple systems as black boxes.

- How well do the other methods perform if images are preprocessed as proposed?

This is an interesting idea, but the purpose of the original benchmarking was to determine how well each method worked “out of the box”. Even so, during our original benchmarking we found that the methods we tested take the filename, not the image, as inputs, so there does not appear to be a straight forward way to determine if our pre-processing method improves the alignment of each method. Finally, having switched to the ANHIR dataset and leaderboard, this experiment does not seem feasible since many of the methods are unavailable.

- How does the RANSAC-based approach compare against other rigid registration methods on low-resolution images?

This too is an interesting question, and to a certain degree is answered by using the ANHIR dataset, as some methods may not take a RANSAC-based approach. However, experiments and answers to this question may be better presented in a separate publication that compares RANSAC and non-RANSAC rigid registration methods.

We would also like to note that while we use RANSAC in the initial feature match filtering, we also use Tukey’s approach and neighbor match filtering to further remove outliers (lines 281-291 in main text). Describing VALIS as a method that removes outliers using RANSAC doesn’t paint the full picture of how VALIS handles feature matching, as RANSAC provides an initial filtering that we refine throughout the pipeline.

In a final application chapter, the authors apply the VALIS toolbox to perform an analysis with multiple markers in the tumor microenvironment. They demonstrate that the software is fit to interdisciplinarily answer a research question. However, this section addresses a question that goes far beyond the alignment of the slides discussed in the first part and already is a small paper in itself. It might be of more value in a separate publication.

While the examples in the application chapter do seem a bit tangential to the software, our motivation is to illustrate how one can use the registered images produced by VALIS to conduct spatial analyses. Our hope is that the publication will be read by a wide audience, some of whom will be more interested in the application than the method itself. Other reviewers have asked to provide additional detailed examples, and so we think keeping the two current examples maintains a good middle ground between keeping the paper focused only on the method and one wherein multiple detailed examples showcase the utility of the method.

I made a few additional comments directly in the manuscript (and apologize if my handwriting might not be legible).

We would have loved to have seen these comments, but unfortunately were not provided with the annotated document.

[redacted]

Reviewer comments, further

Reviewer #1 (Remarks to the Author):

I would like to thank the authors for the work done to revise this manuscript. I will respond to their responses point by point:

- Computing limitations: The authors propose rigid alignment, tiling and micro alignment of tiles. This is an acceptable approach and relevant citations/limitations should be added, along with figures in the manuscript to demonstrate the output of the tiling approach, with multiplexing. The registration error test for the grand challenges is useful but does not seem to be a realistic assessment of micro architectural registration. The ability to align individual nuclei during restaining is a more appropriate test for micro architectural alignment especially for spatial omics applications. The comparison of target registration error at full resolution as compared to lower resolution does not seem realistic as lower resolution should allow for precise macro architectural alignment due to downsampling, placing features closer together and potentially inflating the accuracy. Absolute TRE will also rise due to increasing the image dimensions so some normalization should be made (e.g., maybe % change in TRE relative to % change in spatial dimensions). Either a comparison should be made which evaluates the two in a similar way or significant limitations should be stated. For folks aiming to leverage this tool to study spatial biology, one consideration here would be assessing the change in precision of an effect estimate when increasing the size of the image being registered, e.g., at low resolution, this spatial autocorrelation coefficient is measured less precisely than at high resolution, leading to a more statistically significant effect (something like a volcano plot could be helpful). Image resolution should be stated as a limitation as well since it is still difficult to assess with the presented data.

The inability to use a GPU for registration is a major limitation and should be added to the manuscript.

- Datasets: this is acceptable though the caveat in the limitations should be noted that datasets may exist where VALIS does not exhibit optimal performance but has not been fully explored as such a comprehensive characterization is out of scope. Re: ACROBAT, it would be helpful to publish performance statistics on other datasets in a "Read the Docs" or some website for this package that you can add to as the package is used in other use cases (even community contributions), similar to how other packages illustrate their functionality on novel datasets to encourage usage of the software: e.g., something like https://umap-learn.readthedocs.io/en/latest/exploratory_analysis.html with explored datasets added here e.g., ACROBAT, added to the existing read the docs: <https://valis.readthedocs.io/en/latest/>. That way ACROBAT could be referenced through an external website without requiring explicit publication^[SEP]. Feature description methods: Should also be stated in limitations/future directions^[SEP]. Stain deconvolution: The authors should have some overview figures which demonstrate how this approach visually differs from the prior art. It is unclear how this approach presents an advantage. Some groups may just want to align the nuclear stains to ensure at least the nuclei align to tag cells. The first supplementary figure is helpful but should be elaborated on and compared to a traditional workflow. It should be clear, visually, why this standardization approach is optimal. Stain normalization is a separate issue but should be discussed as well. Perhaps both of these points could be elaborated on in a supplementary discussion section.

- Rewash limitations: This point was addressed, thank you.

- Software: If the authors are unable to post a code ocean capsule, a docker image uploaded to docker hub would suffice, with reference in the read the docs.

- Limitations: Recommendation that they be consolidated or summarized in one section for readability but acceptable as is.

- Species distribution: One thought here could be to move some of the supplementary analyses to the appendix and reference briefly in the text if orthogonal to not detract from the main contributions.^[SEP] - Scope (methods): It can be tough to decide how much to focus on methods/application. Further description of the experimental design and significance of the applications is warranted.^[SEP] - Novelty: I am still unconvinced of novelty in alignment with other reviewers but am respectful of editorial decisions on this matter. Technologies to enhance spatial analyses and facilitate these assessments are needed. One request here would be for the authors to discuss potential application of these technologies for facilitating development of virtual staining algorithms and 3D histology in the discussion / future directions section.

Reviewer #2 (Remarks to the Author):

The authors have addressed most of my concerns. I only have the following minor concerns:
(1) It is still not clear to me how exactly the authors performed the preprocessing. This description "We should clarify that images were resized so that their maximum dimensions were the same" is sufficient for the readers to reproduce the results. Did the authors use the maximum size of a sequence of images or just the maximum of the height and width of each image. Did the authors pad each image to a square? Please provide more details.

(2) Accuracy of ordering the slices. I do not quite understand Fig. d, how can I tell if the algorithm has successfully estimated the slice orders from this figure? Also, please test the ordering algorithm on more datasets, I think this is critical preprocessing step especially for 3D reconstruction.

(3) The resolution of Fig. 7 in the manuscript is quite low. I can barely read texts in Fig. 7.

Reviewer #3 (Remarks to the Author):

The paper describes a toolbox for the registration of WSIs from histopathology. The authors improved their manuscript significantly by adding clarification and additional evaluation. I want to highlight the following points:

1. Due to the many requests by reviewers, the paper is now quite long. I know how frustrating this is, but would it be possible to extract the gist of it and refer to a supplementary section for further details? This would make it much easier to consume.

2. Independently of the structural question, the manuscript could be more concise in many parts. I would encourage to authors to remove redundancies such as lines 167 ff. points 4 and 7 or lines 391 ff and shorten paragraphs across the manuscript.

3. The paper describes a collection of algorithms that - in their combination - outperformed many other algorithms in the field. This has only become evident due to the additional analysis in this revision.

4. In the light of the length of the paper, I would still recommend to move the Application sections to a separate publication, with separate discussion of related work, contributions etc.

I am sorry to read that my additional comments in the document have not been passed on to the authors. I will again try to add them to this revision as some remarks are still applicable.

In addition, here are a few minor remarks:

- The criteria for inclusion in Table 1 is unclear. Why are the methods participating in ANHIR or ACROBAT are not added? Both challenges published the method descriptions of the submitted algorithms.

- The font size in Figure 3 is too small.

- Line 173: "competition"?

Lines 175 ff., I would try to avoid the repetition of "Software that can"

Reviewer #1 (Remarks to the Author)

I would like to thank the authors for the work done to revise this manuscript. I will respond to their responses point by point:

- Computing limitations: The authors propose rigid alignment, tiling and micro alignment of tiles. This is an acceptable approach and relevant citations/limitations should be added, along with figures in the manuscript to demonstrate the output of the tiling approach, with multiplexing.

We are pleased to hear the Reviewer is satisfied with the tiling approach. Regarding figures, the output of the tiling approach is the same as the non-tiling approach, i.e. the generation of deformation field used to non-rigidly warp the images. Examples of these deformation fields are shown in Supplemental Figure 1f. Tiling is only performed to avoid loading the entire image into memory (which may not be possible), but the tiles are eventually stitched together and saved as a large deformation field image (a 2 channel .tif), and the resulting image looks very similar to the non-tiling approach. Also, as single channel images are used for registration, the tiling process is the same for IHC and multiplex IF images. As such, we cannot demonstrate the output of the tiling approach in a way that is specific to multiplexing, as the deformation fields (tiled or non-tiled) is used to warp all channels in the image, whether they are RGB (i.e. single brightfield images) or different antibodies (IF images with multiple channels).

The registration error test for the grand challenges is useful but does not seem to be a realistic assessment of micro architectural registration. The ability to align individual nuclei during restaining is a more appropriate test for micro architectural alignment especially for spatial omics applications.

To our knowledge the ACROBAT and ANHIR grand challenges are the best datasets available to assess registration accuracy. However, we note that the points used to assess registration accuracy in these grand challenges are based on the full resolution images, and include what could be considered “micro architectural features” (see Figure 2 in ANHIR manuscript), although perhaps not nuclear features.

To address the Reviewer’s comment regarding nuclear alignment, we have found a dataset used to assess the accuracy of 3D tissue reconstruction from serial IHC slices, wherein the landmarks are nuclei split by the sectioning blade, and thus found on both images being aligned (Kartasalo et al., 2018). As such, it provides a way to assess how well individual nuclei align, at least for serially sliced brightfield images. Even when using the lower resolution images and no micro-registration, VALIS’ mean target registration error (TRE) of serial images was of 11.4 μ m (N=260 serial slices) (Figure 1a,b below, and Figure 3b,c in main manuscript), an accuracy which exceeds that of the methods tested in the paper (reference score was 15.6 μ m), and that of a recent 3D reconstruction method, CODA, recently published in Nature Methods (Kiemen et al., 2022), which had a mean TRE of 37.1 μ m (as reported in the source data file for Figure 2a). We have added a description this additional benchmarking to the “Benchmarking” subsection.

Our experiments indicate that CyCIF registration tends to be much more accurate than serial IHC (like the Kartasalo dataset) (Figure 5a in main manuscript), which is expected given that the images are the exact same tissue section. CyCIF error was estimated to be around 5 μ m (Figure 5a), but we note that VALIS tends to overestimate error (sometimes up to 16x higher) (Figure 1b,c below, and Figure 3b,c in main manuscript), suggesting that nuclear alignment in CyCIF images should be excellent.

The comparison of target registration error at full resolution as compared to lower resolution does not seem realistic as lower resolution should allow for precise macro architectural alignment due to downsampling, placing features closer together and potentially inflating the accuracy. Absolute TRE will also rise due to increasing the image dimensions so some normalization should be made (e.g., maybe % change in TRE relative to % change in spatial dimensions). Either a comparison should be made which evaluates the two in a similar way or significant limitations should be stated.

The Reviewer makes a good point regarding the TRE reported in Figure 5. This figure shows the estimated error of our internal dataset, which is calculated using the automatically detected and matched image features. As the features were detected on lower resolution images, they do indeed represent macro-architectural features. However, before TRE is calculated, the position of the features is scaled to the full resolution image and converted to biological units, usually μm . As such, the estimated error is for the full resolution image, and because they have meaningful biological units (e.g. μm), can readily be used for comparison without the need for additional normalization. Unfortunately, however, these error estimates are not very accurate, due to the feature positions originating from lower resolution images. However, the error tends to be over-estimated (Figure 1b,c below, and Figure 3b,c in main manuscript), which we demonstrate by comparing VALIS' estimated error to the true error (as estimated using the hand annotated landmarks) using the ACROBAT dataset (results can be downloaded from the leaderboard) and the Kartasalo 2018 dataset. We have added a discussion of this in the "Limitations" section of the manuscript, as well as in the online documentation.

While the Reviewer's comments might apply to the TRE reported with our internal datasets, we note that it is not the case with the TRE scores associated with the grand challenges. The coordinates used to calculate TRE values in the grand challenges are based on the coordinates in the full resolution image, and so the valid concern of comparing TRE at different resolutions described by the Reviewer isn't an issue in these datasets. Additionally, it is only the ANHIR dataset that uses pixel units to calculate TRE (which is normalized to rTRE for comparisons, as suggested by the Reviewer). In contrast, the error in the ACROBAT dataset is in μm , and so all TRE in ACROBAT have the same units, regardless of size of the image used by the registration algorithm. As such, there are no comparisons of TRE values calculated at different resolutions in the grand challenge datasets, as all TRE are calculated using the position of hand matched landmarks found in the full resolution images.

For folks aiming to leverage this tool to study spatial biology, one consideration here would be assessing the change in precision of an effect estimate when increasing the size of the image being registered, e.g., at low resolution, this spatial autocorrelation coefficient is measured less precisely than at high resolution, leading to a more statistically significant effect (something like a volcano plot could be helpful). Image resolution should be stated as a limitation as well since it is still difficult to assess with the presented data.

We note that low resolution images are only used to find the transformation parameters, which are then scaled to warp and save the full resolution image. Therefore, any spatial analyses, including measures of spatial autocorrelation, can be conducted on the registered full resolution image. The resolution of the image used to find the registration parameters should not impact the precision of an effect estimate found during an analysis of the full resolution registered images. As such, VALIS does not limit the user to performing spatial analyses on lower resolution images.

The Reviewer may also be suggesting that using low resolution images to perform the registration will result in less accurate alignments, which in turn may affect the precision of an effect estimate. However,

we would like to point to our experiments that show increasing the size of the image being registered does not always translate into more accurate registrations, and that when there are gains in accuracy, they are often small, as demonstrated with the ANHIR dataset (Supplementary Figure 1a). We would thus argue that differences in effect estimates when using different resolution images to find the registration parameters would also be small.

Given the above, and that the example spatial analyses have been moved to the supplement, we feel that conducting the suggested analysis is outside the scope of the paper.

The inability to use a GPU for registration is a major limitation and should be added to the manuscript.

We have added a *Limitations* section to the manuscript, wherein we mention that VALIS is limited to the CPU.

- Datasets: this is acceptable though the caveat in the limitations should be noted that datasets may exist where VALIS does not exhibit optimal performance but has not been fully explored as such a comprehensive characterization is out of scope.

We are happy to hear that the reviewer finds the datasets acceptable. We have also added a “Limitations” section, wherein we state that VALIS may not work with all datasets.

Re: ACROBAT, it would be helpful to publish performance statistics on other datasets in a “Read the Docs” or some website for this package that you can add to as the package is used in other use cases (even community contributions), similar to how other packages illustrate their functionality on novel datasets to encourage usage of the software: e.g., something like https://umap-learn.readthedocs.io/en/latest/exploratory_analysis.html with explored datasets added here e.g., ACROBAT, added to the existing read the docs: <https://valis.readthedocs.io/en/latest/>. That way ACROBAT could be referenced through an external website without requiring explicit publication.

We thank the Reviewer for this excellent suggestion. We have added a “datasets” page (here) to our online documentation, which now includes ANHIR, ACROBAT, and the 3D datasets presented in (Kartasalo et al., 2018). Regarding the 3D dataset, benchmarking revealed that VALIS scored a mean TRE of 11.4 μ m, which is an improvement over the methods tested in the paper (reference score was 15.6 μ m), and that of a recent 3D reconstruction method recently published in Nature Methods (Kiemen et al., 2022), which had a mean TRE=37.1 μ m (as reported in the source data file for Figure 2a).

- Feature description methods: Should also be stated in limitations/future directions

It is not clear to us what the Reviewer is suggesting is a limitation or what the future direction of the feature description method will be.

- Stain deconvolution: The authors should have some overview figures which demonstrate how this approach visually differs from the prior art. It is unclear how this approach presents an advantage. Some groups may just want to align the nuclear stains to ensure at least the nuclei align to tag cells. The first supplementary figure is helpful but should be elaborated on and compared to a traditional workflow. It should be clear, visually, why this standardization approach is optimal. Stain normalization is a separate issue but should be discussed as well. Perhaps both of these points could be elaborated on in a supplementary discussion section.

VALIS does not rely on stain deconvolution to perform image registration. Likewise, VALIS does not perform, nor rely upon, *stain* normalization. Instead, in the case of brightfield images, VALIS standardizes colorfulness and hue of each image, and then normalizes them such that they have similar distributions of greyscale intensity values. In the case of immunofluorescence, the nuclear channel is used to align images, which would help ensure nuclei align to tag cells.

The goal of this pre-processing method is to make the images appear as similar as possible prior to registration, which should make it easier to identify the matching features. Using stain deconvolution is potentially more difficult, as stain positivity may vary greatly between two images, i.e. using CD8 stain intensity (via stain deconvolution) on one image to align with CK on another image (also via stain deconvolution) would likely not be successful. It is for this reason that VALIS instead uses the whole tissue to perform the alignments, not just the positively stained areas, i.e. those enhanced during stain deconvolution.

As VALIS' focus is on WSI registration, and does not perform stain deconvolution or stain normalization (e.g. (Macenko et al., 2009)), we believe including discussions of these is outside the scope of the manuscript.

- Rewash limitations: This point was addressed, thank you.

- Software: If the authors are unable to post a code ocean capsule, a docker image uploaded to docker hub would suffice, with reference in the read the docs.

This is an excellent suggestion. We have uploaded a Docker image to Docker Hub (<https://hub.docker.com/r/cdgatenbee/valis-wsi>). It's availability and example usage is also described on ReadTheDocs in the 'Installation' section.

- Limitations: Recommendation that they be consolidated or summarized in one section for readability but acceptable as is.

We have added a "Limitations" section to the manuscript, wherein we discuss that VALIS: may not work with all images; has only been tested with brightfield and immunofluorescent images; is limited to the CPU; can be slow with large images; does not precisely estimate registration accuracy.

- Species distribution: One thought here could be to move some of the supplementary analyses to the appendix and reference briefly in the text if orthogonal to not detract from the main contributions.

We agree the analysis section has become more tangential to the main focus of the paper, and have moved it to the supplement.

- Scope (methods): It can be tough to decide how much to focus on methods/application. Further description of the experimental design and significance of the applications is warranted.

The analysis/application section has been moved to the supplement, where it is intended to provide examples of spatial analyses of multiplexed images (generated by VALIS), both when there are highly accurate alignments (CyCIF), and in cases where there may be greater error (serial brightfield images)

analyzed using quadrats). The exact experimental design isn't integral to VALIS' performance (as demonstrated in the grand challenges), and so users of our method will not need to follow the same protocol. Regarding the conclusions, as we only use one illustrative example per application, we cannot draw meaningful conclusions from the results, and so have avoided any discussions of significance.

- Novelty: I am still unconvinced of novelty in alignment with other reviewers but am respectful of editorial decisions on this matter. Technologies to enhance spatial analyses and facilitate these assessments are needed. One request here would be for the authors to discuss potential application of these technologies for facilitating development of virtual staining algorithms and 3D histology in the discussion / future directions section.

We have reduced the "Applications" section to include a brief description of how VALIS can be used to enhance spatial analyses, moving the full analysis to the supplemental section. In addition to showcasing how VALIS can aid in the generation of multiplexed images, we also provide examples of 3D tissue reconstruction and mention that VALIS (and image registration in general), can be used in the development of virtual staining algorithms.

Reviewer #2 (Remarks to the Author)

The authors have addressed most of my concerns. I only have the following minor concerns:

(1) It is still not clear to me how exactly the authors performed the preprocessing. This description "We should clarify that images were resized so that their maximum dimensions were the same" is sufficient for the readers to reproduce the results. Did the authors use the maximum size of a sequence of images or just the maximum of the height and width of each image. Did the authors pad each image to a square? Please provide more details.

We have simplified this resizing step, now each image is rescaled such that its maximum dimension is 850 pixels. In the event that an image has a dimension smaller than 850 pixels, all other images would be rescaled such that the size of their largest dimension is the same as the largest dimension of the smallest image. No image padding was performed, and so all images maintained their original aspect ratio. We have added these details to the "Reading the Slides" subsection of the main manuscript.

(2) Accuracy of ordering the slices. I do not quite understand Fig. d, how can I tell if the algorithm has successfully estimated the slice orders from this figure? Also, please test the ordering algorithm on more datasets, I think this is critical preprocessing step especially for 3D reconstruction.

We agree that the figure was difficult to interpret and have updated it to more clearly show that VALIS can successfully sort images. We now show the images randomly ordered, and then the reconstructed tissue after VALIS has ordered the slices based off feature similarity. We also realized that we neglected to mention that image sorting is optional, and that one can specify the order in which the images are to be aligned. As the Reviewer notes, this is especially critical to 3D reconstruction, which we now provide examples of, both in the paper ("Benchmarking" and Figures 3b,c & 6e) and in the online documentation.

Regarding the testing of 3D datasets, we have now benchmarked using data provided in (Kartasalo et al., 2018) which revealed that VALIS scored a mean TRE of 11.4 μ m, this is an improvement over the methods

tested in the paper (reference score was 15.6 μ m), and that of a 3D reconstruction method, CODA, recently published in Nature Methods (Kiemen et al., 2022), which had mean TRE=37.1 μ m (as reported in the source data file for Figure 2a). Please see Figure 1a,b below, and updated Benchmarking section and Figures 3b,c & 6e in the main manuscript.

(3) The resolution of Fig. 7 in the manuscript is quite low. I can barely read texts in Fig. 7.

We apologize for the poor resolution. We have now moved Fig.7 to the supplement, where it is provided as a high resolution image.

Reviewer #3 (Remarks to the Author)

The paper describes a toolbox for the registration of WSIs from histopathology. The authors improved their manuscript significantly by adding clarification and additional evaluation. I want to highlight the following points:

1. Due to the many requests by reviewers, the paper is now quite long. I know how frustrating this is, but would it be possible to extract the gist of it and refer to a supplementary section for further details? This would make it much easier to consume.

We agree that the paper has indeed become rather lengthy. To alleviate this, we have moved much of the Application section to the supplement.

2. Independently of the structural question, the manuscript could be more concise in many parts. I would encourage to authors to remove redundancies such as lines 167 ff. points 4 and 7 or lines 391 ff and shorten paragraphs across the manuscript.

In addition to moving the Application section, we have edited the manuscript text to make it more digestible. We have also removed point 7 from lines 167 ff, as it was redundant given point 4.

3. The paper describes a collection of algorithms that - in their combination - outperformed many other algorithms in the field. This has only become evident due to the additional analysis in this revision.

4. In the light of the length of the paper, I would still recommend to move the Application sections to a separate publication, with separate discussion of related work, contributions etc.

We agree with the Reviewer, and have moved most of the Application section to the supplement. We now only briefly discuss a few examples to highlight some ways VALIS could be used.

I am sorry to read that my additional comments in the document have not been passed on to the authors. I will again try to add them to this revision as some remarks are still applicable.

Unfortunately, we are still unable to access the comments in the document.

In addition, here are a few minor remarks:

- The criteria for inclusion in Table 1 is unclear. Why are the methods participating in ANHIR or ACROBAT are not added? Both challenges published the method descriptions of the submitted algorithms.

We thank the Reviewer for this excellent point. There are over 60 entries in the combined challenges, and so for the ANHIR challenge we have added the top 6 methods described in the paper (CKVST, TUNI, MEVIS, TUB, AGH/AGHSSO/ DeeperHistReg, UPENN (same as HistoReg, which is already in the table)). Several of the more recent entries for the ANHIR grand challenge appear to be unpublished, although we have also added the published SFG method to the table. We have also included several of the top performers in the ACROBAT challenge that were not part of the ANHIR challenge (AHUAMOA, NEMESIS, MEDAL, GESTALT (same as Marzahl et al. 2021, which is already in the table)).

- The font size in Figure 3 is too small.

We apologize for this and have increased the font size in Figure 3.

- Line 173: "competition"?

We have removed "competition" from the sentence.

Lines 175 ff., I would try to avoid the repetition of "Software that can"

We have reworded this section to reduce the repetition of "Software that can".

Reviewer #1 (Remarks to the Author)

I would like to thank the authors for the work done to revise this manuscript. I will respond to their responses point by point:

- Computing limitations: The authors propose rigid alignment, tiling and micro alignment of tiles. This is an acceptable approach and relevant citations/limitations should be added, along with figures in the manuscript to demonstrate the output of the tiling approach, with multiplexing.

We are pleased to hear the Reviewer is satisfied with the tiling approach. Regarding figures, the output of the tiling approach is the same as the non-tiling approach, i.e. the generation of deformation field used to non-rigidly warp the images. Examples of these deformation fields are shown in Supplemental Figure 2d. Tiling is only performed to avoid loading the entire image into memory (which may not be possible), but the tiles are eventually stitched together and saved as a large deformation field image (a 2 channel .tif), and the resulting image looks very similar to the non-tiling approach. As single channel images are used for registration, the tiling process is the same for IHC and multiplex IF images. As such, we cannot demonstrate the output of the tiling approach in a way that is specific to multiplexing, as the deformation fields (tiled or non-tiled) is used to warp all channels in the image, whether they are RGB (i.e. single brightfield images) or different antibodies (IF images with multiple channels).

The registration error test for the grand challenges is useful but does not seem to be a realistic assessment of micro architectural registration. The ability to align individual nuclei during restaining is a more appropriate test for micro architectural alignment especially for spatial omics applications.

To our knowledge the ACROBAT and ANHIR grand challenges are the best datasets available to assess registration accuracy. However, we note that the points used to assess registration accuracy in these grand challenges are based on the full resolution images, and include what could be considered “micro architectural features” (see Figure 2 in ANHIR manuscript), although perhaps not nuclear features.

To address the Reviewer’s comment regarding nuclear alignment, we have found a dataset used to assess the accuracy of 3D tissue reconstruction from serial IHC slices, wherein the landmarks are nuclei split by the sectioning blade, and thus found on both images being aligned (Kartasalo et al., 2018). As such, it provides a way to assess how well individual nuclei align, at least for serially sliced brightfield images. Even when using the lower resolution images and no micro-registration, VALIS’ mean target registration error (TRE) of serial images was of 11.4 μ m (N=260 serial slices) (Figure 1a,b below, and Figure 3b,c in main manuscript), an accuracy which exceeds that of the methods tested in the paper (reference score was 15.6 μ m), and that of a recent 3D reconstruction method, CODA, recently published in Nature Methods (Kiemien et al., 2022), which had a mean TRE of 37.1 μ m (as reported in the source data file for Figure 2a). We have added a description this additional benchmarking to the “Benchmarking” subsection.

Our experiments indicate that CyCIF registration tends to be much more accurate than serial IHC (like the Kartasalo dataset) (Figure 5a in main manuscript), which is expected given that the images are the exact same tissue section. CyCIF error was estimated to be around 5 μ m (Figure 5a), but we note that VALIS tends to overestimate error (sometimes up to 16x higher) (Figure 1b,c below, and Figure 3b,c in main manuscript), suggesting that nuclear alignment in CyCIF images should be excellent.

The comparison of target registration error at full resolution as compared to lower resolution does not seem realistic as lower resolution should allow for precise macro architectural alignment due to downsampling, placing features closer together and potentially inflating the accuracy. Absolute TRE will also rise due to increasing the image dimensions so some normalization should be made (e.g., maybe % change in TRE relative to % change in spatial dimensions). Either a comparison should be made which evaluates the two in a similar way or significant limitations should be stated.

The Reviewer makes a good point regarding the TRE reported in Figure 5. This figure shows the estimated error of our internal dataset, which is calculated using the automatically detected and matched image features. As the features were detected on lower resolution images, they do indeed represent macro-architectural features. However, before TRE is calculated, the position of the features is scaled to the full resolution image and converted to biological units, usually μm . As such, the estimated error is for the full resolution image, and because they have meaningful biological units (e.g. μm), can readily be used for comparison without the need for additional normalization. Unfortunately, however, these error estimates are not very accurate, due to the feature positions originating from lower resolution images. However, the error tends to be over-estimated (Figure 1b,c below, and Figure 3b,c in main manuscript), which we demonstrate by comparing VALIS' estimated error to the true error (as estimated using the hand annotated landmarks) using the ACROBAT dataset (results can be downloaded from the leaderboard) and the Kartasalo 2018 dataset. We have added a discussion of this in the "Limitations" section of the manuscript, as well as in the online documentation.

While the Reviewer's comments might apply to the TRE reported with our internal datasets, we note that it is not the case with the TRE scores associated with the grand challenges. The coordinates used to calculate TRE values in the grand challenges are based on the coordinates in the full resolution image, and so the valid concern of comparing TRE at different resolutions described by the Reviewer isn't an issue in these datasets. Additionally, it is only the ANHIR dataset that uses pixel units to calculate TRE (which is normalized to rTRE for comparisons, as suggested by the Reviewer). In contrast, the error in the ACROBAT dataset is in μm , and so all TRE in ACROBAT have the same units, regardless of size of the image used by the registration algorithm. As such, there are no comparisons of TRE values calculated at different resolutions in the grand challenge datasets, as all TRE are calculated using the position of hand matched landmarks found in the full resolution images.

For folks aiming to leverage this tool to study spatial biology, one consideration here would be assessing the change in precision of an effect estimate when increasing the size of the image being registered, e.g., at low resolution, this spatial autocorrelation coefficient is measured less precisely than at high resolution, leading to a more statistically significant effect (something like a volcano plot could be helpful). Image resolution should be stated as a limitation as well since it is still difficult to assess with the presented data.

We note that low resolution images are only used to find the transformation parameters, which are then scaled to warp and save the full resolution image. Therefore, any spatial analyses, including measures of spatial autocorrelation, can be conducted on the registered full resolution image. The resolution of the image used to find the registration parameters should not impact the precision of an effect estimate found during an analysis of the full resolution registered images. As such, VALIS does not limit the user to performing spatial analyses on lower resolution images.

The Reviewer may also be suggesting that using low resolution images to perform the registration will result in less accurate alignments, which in turn may affect the precision of an effect estimate. However,

we would like to point to our experiments that show increasing the size of the image being registered does not always translate into more accurate registrations, and that when there are gains in accuracy, they are often small, as demonstrated with the ANHIR dataset (Supplementary Figure 1a). We would thus argue that differences in effect estimates when using different resolution images to find the registration parameters would also be small.

Given the above, and that the example spatial analyses have been moved to the supplement, we feel that conducting the suggested analysis is outside the scope of the paper.

The inability to use a GPU for registration is a major limitation and should be added to the manuscript.

We have added a *Limitations* section to the manuscript, wherein we mention that VALIS is limited to the CPU.

- Datasets: this is acceptable though the caveat in the limitations should be noted that datasets may exist where VALIS does not exhibit optimal performance but has not been fully explored as such a comprehensive characterization is out of scope.

We are happy to hear that the reviewer finds the datasets acceptable. We have also added a *Limitations* section, wherein we state that VALIS may not work with all datasets.

Re: ACROBAT, it would be helpful to publish performance statistics on other datasets in a “Read the Docs” or some website for this package that you can add to as the package is used in other use cases (even community contributions), similar to how other packages illustrate their functionality on novel datasets to encourage usage of the software: e.g., something like https://umap-learn.readthedocs.io/en/latest/exploratory_analysis.html with explored datasets added here e.g., ACROBAT, added to the existing read the docs: <https://valis.readthedocs.io/en/latest/>. That way ACROBAT could be referenced through an external website without requiring explicit publication.

We thank the Reviewer for this excellent suggestion. We have added a “Datasets” page (here) to our online documentation, which now includes ANHIR, ACROBAT, and the 3D datasets presented in (Kartasalo et al., 2018). Regarding the 3D dataset, benchmarking revealed that VALIS scored a mean TRE of 11.4 μ m, which is an improvement over the methods tested in the paper (reference score was 15.6 μ m), and that of a recent 3D reconstruction method recently published in Nature Methods (Kiemen et al., 2022), which had a mean TRE=37.1 μ m (as reported in the source data file for Figure 2a).

- Feature description methods: Should also be stated in limitations/future directions

It is not clear to us what the Reviewer is suggesting is a limitation or what the future direction of the feature description method will be.

- Stain deconvolution: The authors should have some overview figures which demonstrate how this approach visually differs from the prior art. It is unclear how this approach presents an advantage. Some groups may just want to align the nuclear stains to ensure at least the nuclei align to tag cells. The first supplementary figure is helpful but should be elaborated on and compared to a traditional workflow. It should be clear, visually, why this standardization approach is optimal. Stain normalization is a separate issue but should be discussed as well. Perhaps both of these points could be elaborated on in a supplementary discussion section.

VALIS does not rely on stain deconvolution to perform image registration. Likewise, VALIS does not perform, nor rely upon, *stain* normalization. Instead, in the case of brightfield images, VALIS standardizes colorfulness and hue of each image, and then normalizes them such that they have similar distributions of intensity values. We have added benchmarking to demonstrate that this approach yields more accurate results than other commonly used preprocessing methods, such as grayscale conversion, global histogram equalization, and contrast limited histogram equalization (Figure 1e below, Supplemental Figure 1b in the main manuscript). In the case of immunofluorescence, the nuclear channel is used to align images, which would help ensure nuclei align to tag cells.

The goal of our brightfield pre-processing method is to make the images appear as similar as possible prior to registration, which should make it easier to identify the matching features. Using stain deconvolution is potentially more difficult, as stain positivity may vary greatly between two images, i.e. using CD8 stain intensity (via stain deconvolution) on one image to align with CK on another image (also via stain deconvolution) would likely not be successful. It is for this reason that VALIS instead uses the whole tissue to perform the alignments, not just the positively stained areas, i.e. those enhanced during stain deconvolution.

As VALIS' focus is on WSI registration, and does not perform stain deconvolution or stain normalization (e.g. (Macenko et al., 2009)), we believe including discussions of these topics is outside the scope of the manuscript.

- Rewash limitations: This point was addressed, thank you.

- Software: If the authors are unable to post a code ocean capsule, a docker image uploaded to docker hub would suffice, with reference in the read the docs.

This is an excellent suggestion. We have uploaded a Docker image to Docker Hub (<https://hub.docker.com/r/cdgatenbee/valis-wsi>). It's availability and example usage is also described on ReadTheDocs in the 'Installation' section.

- Limitations: Recommendation that they be consolidated or summarized in one section for readability but acceptable as is.

We have added a *Limitations* section to the manuscript, wherein we discuss that VALIS: may not work with all images; has only been tested with brightfield and immunofluorescent images; is limited to the CPU; can be slow with large images; does not precisely estimate registration accuracy.

- Species distribution: One thought here could be to move some of the supplementary analyses to the appendix and reference briefly in the text if orthogonal to not detract from the main contributions.

We agree the analysis section has become more tangential to the main focus of the paper, and have moved it to the supplement.

- Scope (methods): It can be tough to decide how much to focus on methods/application. Further description of the experimental design and significance of the applications is warranted.

The analysis/application section has been moved to the supplement, where it is intended to provide examples of spatial analyses of multiplexed images (generated by VALIS), both when there are highly accurate alignments (CyCIF), and in cases where there may be greater error (serial brightfield images analyzed using quadrats). The exact experimental design isn't integral to VALIS' performance (as demonstrated in the grand challenges), and so users of our method will not need to follow the same protocol. Regarding the conclusions, as we only use one illustrative example per application, we cannot draw meaningful conclusions from the results, and so have avoided any discussions of significance.

- Novelty: I am still unconvinced of novelty in alignment with other reviewers but am respectful of editorial decisions on this matter. Technologies to enhance spatial analyses and facilitate these assessments are needed. One request here would be for the authors to discuss potential application of these technologies for facilitating development of virtual staining algorithms and 3D histology in the discussion / future directions section.

We have reduced the *Applications* section to include a brief description of how VALIS can be used to enhance spatial analyses, moving the full analysis to the supplemental section. In addition to showcasing how VALIS can aid in the generation of multiplexed images, we also provide examples of 3D tissue reconstruction and mention that VALIS (and image registration in general), can be used in the development of virtual staining algorithms.

Reviewer #2 (Remarks to the Author)

The authors have addressed most of my concerns. I only have the following minor concerns:

(1) It is still not clear to me how exactly the authors performed the preprocessing. This description "We should clarify that images were resized so that their maximum dimensions were the same" is sufficient for the readers to reproduce the results. Did the authors use the maximum size of a sequence of images or just the maximum of the height and width of each image. Did the authors pad each image to a square? Please provide more details.

We have simplified this resizing step, now each image is rescaled such that its maximum dimension is 850 pixels. In the event that an image has a dimension smaller than 850 pixels, all other images would be rescaled such that the size of their largest dimension is the same as the largest dimension of the smallest image. No image padding was performed, and so all images maintained their original aspect ratio. We have added these details to the "Reading the Slides" subsection of the main manuscript.

(2) Accuracy of ordering the slices. I do not quite understand Fig. d, how can I tell if the algorithm has successfully estimated the slice orders from this figure? Also, please test the ordering algorithm on more datasets, I think this is critical preprocessing step especially for 3D reconstruction.

We agree that the figure was difficult to interpret and have updated it to more clearly show that VALIS can successfully sort images. We now show the images randomly ordered, and then the reconstructed tissue after VALIS has ordered the slices based off feature similarity. We also realized that we neglected to mention that image sorting is optional, and that one can specify the order in which the images are to be aligned. As the Reviewer notes, this is especially critical to 3D reconstruction, which we now provide examples of, both in the paper ("Benchmarking" and Figures 3b,c & 6e) and in the online documentation.

Regarding the testing of 3D datasets, we have now benchmarked using data provided in (Kartasalo et al., 2018) which revealed that VALIS scored a mean TRE of 11.4 μ m, this is an improvement over the methods tested in the paper (reference score was 15.6 μ m), and that of a 3D reconstruction method, CODA, recently published in Nature Methods (Kiemen et al., 2022), which had mean TRE=37.1 μ m (as reported in the source data file for Figure 2a). Please see Figure 1a,b below, and updated Benchmarking section and Figures 3b,c & 6e in the main manuscript.

(3) The resolution of Fig. 7 in the manuscript is quite low. I can barely read texts in Fig. 7.

We apologize for the poor resolution. We have now moved Fig.7 to the supplement, where it is provided as a high resolution image.

Reviewer #3 (Remarks to the Author)

The paper describes a toolbox for the registration of WSIs from histopathology. The authors improved their manuscript significantly by adding clarification and additional evaluation. I want to highlight the following points:

1. Due to the many requests by reviewers, the paper is now quite long. I know how frustrating this is, but would it be possible to extract the gist of it and refer to a supplementary section for further details? This would make it much easier to consume.

We agree that the paper has indeed become rather lengthy. To alleviate this, we have moved much of the Application section to the supplement.

2. Independently of the structural question, the manuscript could be more concise in many parts. I would encourage to authors to remove redundancies such as lines 167 ff. points 4 and 7 or lines 391 ff and shorten paragraphs across the manuscript.

In addition to moving the Application section, we have edited the manuscript text to make it more digestible. We have also removed point 7 from lines 167 ff, as it was redundant given point 4.

3. The paper describes a collection of algorithms that - in their combination - outperformed many other algorithms in the field. This has only become evident due to the additional analysis in this revision.

4. In the light of the length of the paper, I would still recommend to move the Application sections to a separate publication, with separate discussion of related work, contributions etc.

We agree with the Reviewer, and have moved most of the Application section to the supplement. We now only briefly discuss a few examples to highlight some ways VALIS could be used.

In addition, here are a few minor remarks:

- The criteria for inclusion in Table 1 is unclear. Why are the methods participating in ANHIR or ACROBAT are not added? Both challenges published the method descriptions of the submitted algorithms.

We thank the Reviewer for this excellent point. There are over 60 entries in the combined challenges, and so for the ANHIR challenge we have added the top 6 methods described in the paper (CKVST, TUNI, MEVIS, TUB, AGH/AGHSSO/ DeeperHistReg, UPENN (same as HistoReg, which is already in the table)). Several of the more recent entries for the ANHIR grand challenge appear to be unpublished, although we have also added the published SFG method to the table. We have also included several of the top performers in the ACROBAT challenge that were not part of the ANHIR challenge (AHUAMOA, NEMESIS, MEDAL, GESTALT (same as Marzahl et al. 2021, which is already in the table)).

- The font size in Figure 3 is too small.

We apologize for this and have increased the font size in Figure 3.

- Line 173: "competition"?

We have removed "competition" from the sentence.

- Lines 175 ff., I would try to avoid the repetition of "Software that can"

We have reworded this section to reduce the repetition of "Software that can".

-I am sorry to read that my additional comments in the document have not been passed on to the authors. I will again try to add them to this revision as some remarks are still applicable.

We have now had the opportunity to see the additional comments in the document, and address each below:

1. This statement needs more backup by the experiments conducted or a weaker formulation (regarding highlighted text in the Abstract: "Benchmarking indicates that VALIS is also the most accurate and robust method currently available.").

We have replaced this statement with one describing VALIS' performance in the ANHIR challenge and 3D reconstruction benchmarking.

2. "I would suggest to mention that it is built upon existing OSS methods" (regarding highlighted text in the Abstract, "VALIS is written in Python and requires only few lines of code for execution").

We agree it would be good to emphasize VALIS is built upon existing opensource software tools, and have updated the abstract accordingly.

3. “One of two? Or each analysis?” (regarding the two example spatial analyses described in the Abstract).

As the example analyses have been moved to the supplement, we have removed this portion of the abstract.

4. This is saying you can rarely do successful image registration in pathology? (regarding highlighted text on page 4: “However, successful image registration requires that the images look similar, but this requirement is rarely satisfied”).

Our intention here was to present the challenges associated with registering pathology images. We certainly don’t want to suggest that image registration in pathology is rarely successful, and so have updated the statement to “However, aligning histology images presents several challenges, which include:...”.

5. What do you mean “similar”? (in reference to the same text highlighted in comment 4).

We have replaced the text in reference, and no longer state that image registration requires that images look similar.

6. Restructure? (in reference to description of a recent image registration pipeline described in Chiaruttini et al., 2022).

The discussion of this manual pipeline was removed to keep the paper more focused on automated methods.

7. Maybe add some of the methods that were successful in the ANHIR challenge? (in reference to the highlighted text on page 4 “Until only recently, many methods have been missing features that would make them suitable for large datasets of multi-gigabyte whole slide images”).

We have updated the manuscript, and Table 1, to include some more of the methods from the ANHIR challenge.

8. ANHIR participants? Maybe also add mmTRE from ANHIR where available? (In reference to Table 1).

We have updated Table 1 to include more of the methods from the ANHIR challenge. We have also added a MMrTRE column and provided values where available, using the values on the leaderboard (Feb 20, 2023).

9. Insert “to” (highlighted text on page 6).

We thank the Reviewer for catching this typo. We have reworked this section of the paper, and have fixed this.

10. How long does conversion to vips take (in reference to highlighted text on page 7 “then combine the tiles to rebuild the entire image as single whole-slide libvips image”).

We have timed the conversion from WSI to a vips image, with the results shown in Figure 1d below and Supplemental Figure 1a in the main text. Conversion of smaller images, e.g. those extracted from the image pyramid, takes seconds. As smaller images are used in the registration pipeline, this conversion does not have a significant impact on registration speed. In the case of larger images (e.g. 40,000 x 40,000), the conversion may take over a minute, but this only occurs during the final warping and saving of the full resolution image, which may or may not be utilized depending on the application.

11. How does this compare to the state of the art? (in reference to highlighted text on page 8 “The default method in VALIS is to standardize the color information from the image”).

This is an excellent question. To answer, we compared registration error using VALIS’ default pre-processing method with results using preprocessing methods frequently used in the ANHIR challenge. Specifically, we compared our preprocessing method to grayscale conversion, global histogram equalization of grayscale images, and contrast limited adaptive histogram equalization (CLAHE) of grayscale images. Benchmarking reveals that our preprocessing method produces registrations with relatively low mean error and small dispersion, indicating it yields more accurate and robust registration results than other frequently used methods (Figure 1e below, and Supplemental Figure 1b in the main text).

Stain deconvolution was another popular preprocessing method in the ANHIR challenge, but we have purposefully avoided making stain deconvolution the default method because it requires a stain matrix, which either needs to be provided by the user or estimated by the software (which can be challenging). Furthermore, color deconvolution highlights the differing spatial distribution of stains, which can make registration more challenging, unless one is only working a single stain such as H&E.

12. Clarify if “novel” refers to the alignment or the sorting. (in reference to highlighted text on page 8 “provides a novel pipeline”).

We have removed the use of novel here, and instead described VALIS as providing a groupwise registration method.

13. I would propose to apply VALIS to the ongoing ANHIR challenge (in reference to highlighted text on page 11 “how well does VALIS performs in comparison”).

We again would like to thank the Reviewer for this suggestion, which was addressed in the previous round of reviews.

14. Isn’t this a bit too much? (in reference to highlighted text on page 12, “A unique and powerful feature of VALIS...”)

We agree that this was indeed too much and have removed this phrase.

15. In my experience, the order of the slides is known in most cases or it’s dictated by the application (e.g. register everything to H&E). (in reference to the discussion on page 12 about the benefits of not needing to define a reference image).

We can only speak from our experience, which differs from that of the Reviewer. In our case, we developed VALIS because we were working on a project where we needed to register dual-stained serial IHC slices, none of which were H&E, nor numbered according to slicing order (Gatenbee et al. 2022). In this project, there did not seem to be a single stain that consistently worked as the reference image, and because we did not know the slice order, there was no obvious way to decide which pairs of images should be aligned (e.g. aligning in slice order). This led us to develop a method to sort the images based on similarity, meaning that we did not need an H&E image, nor know the order in which the tissue was cut. While this scenario may not be the norm, we believe this additional flexibility (i.e. not needing a reference H&E image or knowing slice order) is a useful feature, and it has come in handy in several of our other projects where we do not know the slice order nor have the H&E image. We would like to note, however, that if this information is available, it can be used by VALIS. For example, the image order can be provided if performing 3D reconstruction, or a reference image can be specified if one wants to align everything directly it.

16. I can follow this argumentation and given the large effort it takes to create manual landmarks, it might make sense to do it this way. But still, it leaves kind of a bad taste that the features created for the winning method are used to evaluate its comparative accuracy. (in reference to highlighted text on page 15, "In order to quantify the registration error, we calculated the median distance (in μm) between matched features after alignment").

We agree with the Reviewer that the original benchmarking was not ideal. In our previous response we performed benchmarking using the ANHIR and ACROBAT grand challenges, which removes the bias associated with using the automatically found landmarks. In this response, we have added the 3D reconstruction benchmarking datasets as well. Together, we hope that VALIS' performance with these three datasets will be more convincing than the initial benchmarking.

Reviewer comments, final round review

Reviewer #1 (Remarks to the Author):

The authors have addressed most of the reviewer concerns.

Reviewer #2 (Remarks to the Author):

The authors have successfully addressed all my concerns. I do not have any additional comments.

Reviewer #3 (Remarks to the Author):

I want to thank the authors, I have no further comments. All concerns have been addressed."